# CSFV restricts necroptosis to sustain infection by inducing autophagy/mitophagy-targeted degradation of RIPK3

Keke Wu,[1,2] Bingke Li,[1,2] Xiaoai Zhang,[2] Yiqi Fang,[1,3] Sen Zeng,[1,3] Wenshuo Hu,[1,3] Xiaodi Liu,[1,3] Xueyi Liu,[1,3] Zhimin Lu,[1,3] Xiaowen Li,[1,3] Wenxian Chen,[1,3] Yuwei Qin,[1,3] Bolun Zhou,[1,3] Linke Zou,[1,3] Feifan Zhao,[1,3] Lin Yi,[1,2,3] Mingqiu Zhao,[1,2,3] Shuangqi Fan,[1,2,3] Jinding Chen[1,2,3]

**ABSTRACT**  As an essential component of host defense against infection, necroptosis is a novel highly regulated mode of cell death that is mediated by signaling complexes containing receptor-interacting protein kinase 1 (RIPK1) and RIPK3. Lymphocyte depletion and immunosuppression are typical clinical features of pigs infected with classical swine fever virus (CSFV). Here, we provide the first evidence for the involvement of necroptosis in the necrosis of T lymphocytes in the spleen and peripheral blood of pigs infected *in vivo* with CSFV. However, it is true for some viruses with non-cytopathic effects, including CSFV, to balance host defense against infection. *In vitro*, the induction of autophagy by CSFV at a later stage of infection clearly restricts necroptosis. Mechanistic studies revealed that CSFV NS4A protein promoted tripartite motif-containing 25 expression, synergistically induced the occurrence of mitophagy, targeted the autophagic degradation of RIPK3 to block the progression of necroptosis occurrence, and achieved persistent viral infection. Interestingly, we found that RIPK3 was able to specifically localize at the outer mitochondrial membrane, and the autophagy receptor NDP52 was most likely involved in the autophagic degradation of RIPK3 during CSFV infection. Our findings provide evidence supporting that the CSFV-induced autophagy pathway plays an important role in counteracting host cell necrosis, enriching our knowledge of pathogens that may subvert and evade necroptosis this host defense, and shedding new light on understanding the mechanisms of T lymphocyte exhaustion and immunosuppression during CSFV infection.

**IMPORTANCE**  CSFV infection in pigs causes persistent high fever, hemorrhagic necrotizing multi-organ inflammation, and high mortality, which seriously threatens the global swine industry. Cell death is an essential immune response of the host against pathogen invasion, and lymphopenia is the most typical clinical feature in the acute phase of CSFV infection, which affects the initial host antiviral immunity. As an "old" virus, CSFV has evolved mechanisms to evade host immune response after a long genetic evolution. Here, we show that necroptosis is a limiting host factor for CSFV infection and that CSFV-induced autophagy can subvert this host defense mechanism to promote its sustained replication. Our findings reveal a complex link between necroptosis and autophagy in the process of cell death, provide evidence supporting the important role for CSFV in counteracting host cell necrosis, and enrich our knowledge of pathogens that may subvert and evade this host defense.

**KEYWORDS**  autophagy, necroptosis, CSFV, NS4A, TRIM25, mitophagy, autophagic degradation, RIPK3

N ecroptosis (also known as cell-programmed necroptosis) emerges as a novel signaling molecule-regulated, orderly manner of cell death that retains some

Address correspondence to Jinding Chen, jdchen@scau.edu.cn.

The authors declare no conflict of interest.

See the funding table on p. 27.

features of apoptosis, such as being genetically controlled, and exhibits necrosis-like morphological features, such as inflammatory responses that can cause surrounding tissues (1, 2). In the absence of apoptotic conditions (mainly when caspase 8 is inhibited), necroptosis can be triggered by multiple signaling pathways, including the tumor necrosis factor receptor family (TNFR), the Toll-like family of receptors, and the cytoplasmic DNA sensor Z-DNA-binding protein 1 (ZBP1/DAI), among others (3–5). Receptor-interacting protein kinase 1 (RIPK1) and RIPK3 form a protein complex called "necrosome" through their respective RIP homotypic interaction motif (RHIM) domains, activated RIPK3 recruits and activates mixed lineage kinase domain-like pseudokinase (MLKL) (6), and the oligomerized MLKL translocates from the cytoplasm to the cell membrane and organelle membranes to form a permeability pore, causing cell necrosis (7, 8).

As a component of host defense against infection, necroptosis is closely related to the body's innate immune response, the development and progression of inflammation-related diseases, and so on (9–11). This contribution was first derived from the study of vaccinia virus (VV), in which mice were infected with VV and specifically induced the formation of an RIPK1-RIPK3 necrosome complex in the liver, triggering tissue necroptosis and inflammation, leading to elevated viral replication and mouse mortality (10). Subsequently, it has been documented that upon infection of host cells with influenza A virus (IAV), ZBP1 recognizes the IAV genome and in turn activates host RIPK3 kinase, which induces cell death through the MLKL-driven necroptosis pathway (12, 13). In addition, bovine parvovirus, murine cytomegalovirus, and reovirus, among others, can activate RIPK3-dependent necroptosis (14, 15). The spread of the virus depends on the state of the infected cell, as well as the interactions between the virus and eukaryotic cells. Cells die after infection with the virus and prevent virus propagation, whereas some viruses encode suppressor genes of cell death to counteract this protective mechanism (16–18). Herpes simplex virus type 1 (HSV1) depends on the cognate RHIM suppressor to prevent ZBP1 activation by newly synthesized RNA in human cells (19). In addition, Epstein-Barr virus-encoded LMP1 promotes K63-linked polyubiquitination of RIPK1 and inhibits protein expression, while inhibiting K63-linked polyubiquitination of RIPK3 prevents necroptosis in human nasopharyngeal epithelial and nasopharyngeal carcinoma cells (20). Necroptosis is important for the innate immune control of virus-induced inflammation and viral infection, while correspondingly, viruses have also evolved counterpart strategies to evade the host immune response (21).

It is generally accepted that active cell death modalities mainly include apoptosis, necroptosis, and autophagy. As a highly conserved intracellular metabolic process in eukaryotes, autophagy delivers damaged organelles, misfolded or aggregated proteins, and invading pathogens to lysosomes for degradation or recycling (22). Selective trafficking of autophagy substrates requires multiple cargo receptor mediation, including sequestosome-1 (SQSTM1/P62), nuclear dot protein 52 (CALCOCO2/NDP52), optineurin (OPTN), neighbor of BRCA1 gene 1 (NBR1), Tax1-binding protein 1 (TAX1BP1), Toll-interacting protein (TOLLIP), etc. (23–25). Autophagy receptors, which recognize a variety of substrates (damaged organelles such as mitochondria, ubiquitinated or non-ubiquitinated proteins) and directly bind microtubule-associated protein 1 light chain 3 (MAP1LC3/LC3) for lysosomal degradation, are hubs linking autophagy substrates to lysosomes (22, 26). Both autophagy and necroptosis are highly regulated and both balance cell death and survival, and although several reports have suggested reciprocal crosstalk and overlap between them (27–29), to date the mechanistic details remain poorly understood.

Classical swine fever virus (CSFV), which is the same *Flaviviridae* as hepatitis C and dengue viruses, is an enveloped RNA virus and the etiological agent of swine fever (CSF) (30). The mature CSFV particle structure includes four structural proteins [core protein (C) and envelope glycoprotein (E$^{rns}$, E1, and E2)] and eight non-structural proteins (N$^{pro}$, P7, NS2, NS3, NS4A, NS4B, NS5A, and NS5B) (31–37). CSFV infection of pigs causes persistent high fever, hemorrhagic necrotizing multi-organ inflammation with high mortality, and

seriously threatens the global swine industry. As an "old" virus, CSFV has undergone a long genetic evolution and has evolved related mechanisms to escape the host immune response (38, 39). All leukocyte populations in peripheral blood, especially lymphopenia, are the most typical clinical features in the acute phase of CSFV infection, thereby affecting the initial host antiviral immunity. Furthermore, interestingly, when CSFV was propagated in host cells *in vitro*, there was no obvious cytopathic effect. Although many studies related to CSFV replication have been performed, the mechanism of viral escape from this disease is unknown.

Previously, we have found that CSFV can induce autophagy and utilize autophagic vesicles for replication (40, 41), and furthermore, CSFV was able to induce mitochondrial fission and mitophagy to promote its infection process (42). In this study, we found that CSFV infection *in vivo* was able to activate necroptosis in peripheral blood mononuclear cells (PBMCs) and spleen, leading to lymphocyte necrosis. Whereas, in contrast to this behavior, CSFV clearly inhibited necroptosis at a later stage of infection *in vitro*. Further mechanistic studies revealed that the CSFV NS4A protein interacts with the E3 ubiquitin ligase tripartite-motif containing 25 (TRIM25) (43, 44), promoted its expression, and targeted it to mediate autophagic degradation of RIPK3 and thereby blocked the progression of necroptosis genesis, achieving persistent viral infection. Interestingly, we found that RIPK3 was able to specifically localize at the outer mitochondrial membrane, and the autophagy receptor NDP52 was most likely involved in the autophagic degradation of RIPK3 during CSFV infection. Our results provide evidence supporting an important role for CSFV NS4A in counteracting host cell necrosis, enrich our knowledge of pathogens that may subvert and evade this host defense, and shed new light on understanding the mechanisms of T lymphocyte exhaustion and immunosuppression during CSFV infection.

## RESULTS

### Peripheral blood T lymphopenia caused by CSFV infection of piglets was associated with necroptosis

CSFV has a strong tropism for epithelial cells and immune cells and is a typical immunosuppressive virus that seriously damages the hematopoietic and immune systems (45). First, we investigated the effect of CSFV infection on the number of lymphocytes and their subsets, and we found that the proportion of lymphocytes among peripheral blood mononuclear cells in infected piglets transiently rose to 73.33% at 1 dpi from 65.44% before the challenge and decreased to 6.00% at 6 dpi, where CD3$^+$ and CD4$^+$ T lymphocytes decreased to 2.53% and 1.33% from 19.86% and 4.15% before the challenge, respectively (Fig. 1A and B). Experimental piglets were sacrificed at 7 dpi for pathological examination and specimen collection. We further observed the histopathological changes in the spleen, and the red and white pulp of the spleen were well demarcated in uninfected control pigs, the splenic corpuscles were structurally intact and regularly shaped, and the central artery was clearly visible (black arrow).

Swelling of splenic trabeculae and more rounded cavities (black arrows) in the cytoplasm of smooth muscle cells were observed in the CSFV-infected group of pigs. Splenic corpuscles were sparse and small in size, the boundaries of most splenic corpuscles were already indistinct, and splenic marginal zone and red pulp hemorrhages were severe. Lymphocytes are greatly reduced in number and lightly pigmented. The wall of the splenic artery showed hyalinosis, with thickening and homogeneous red staining of the wall and compaction of the lumen (yellow arrow). Focal disintegration of lymphocytic necrosis with infiltration of granulocytes around necrotic cells is seen in the white pulp (black dotted circle). In addition, the splenic sinusoids were markedly dilated and visible with an excess of packed red blood cells (red arrows) (Fig. 1C). Consistent with previous reports, CSFV infection of spleens induced the development of severe hemorrhagic, necroinflammation (46). These data suggest that CSFV infection caused extensive lymphocyte depletion and tissue necrosis; however, the underlying mechanism was unclear and controversial.

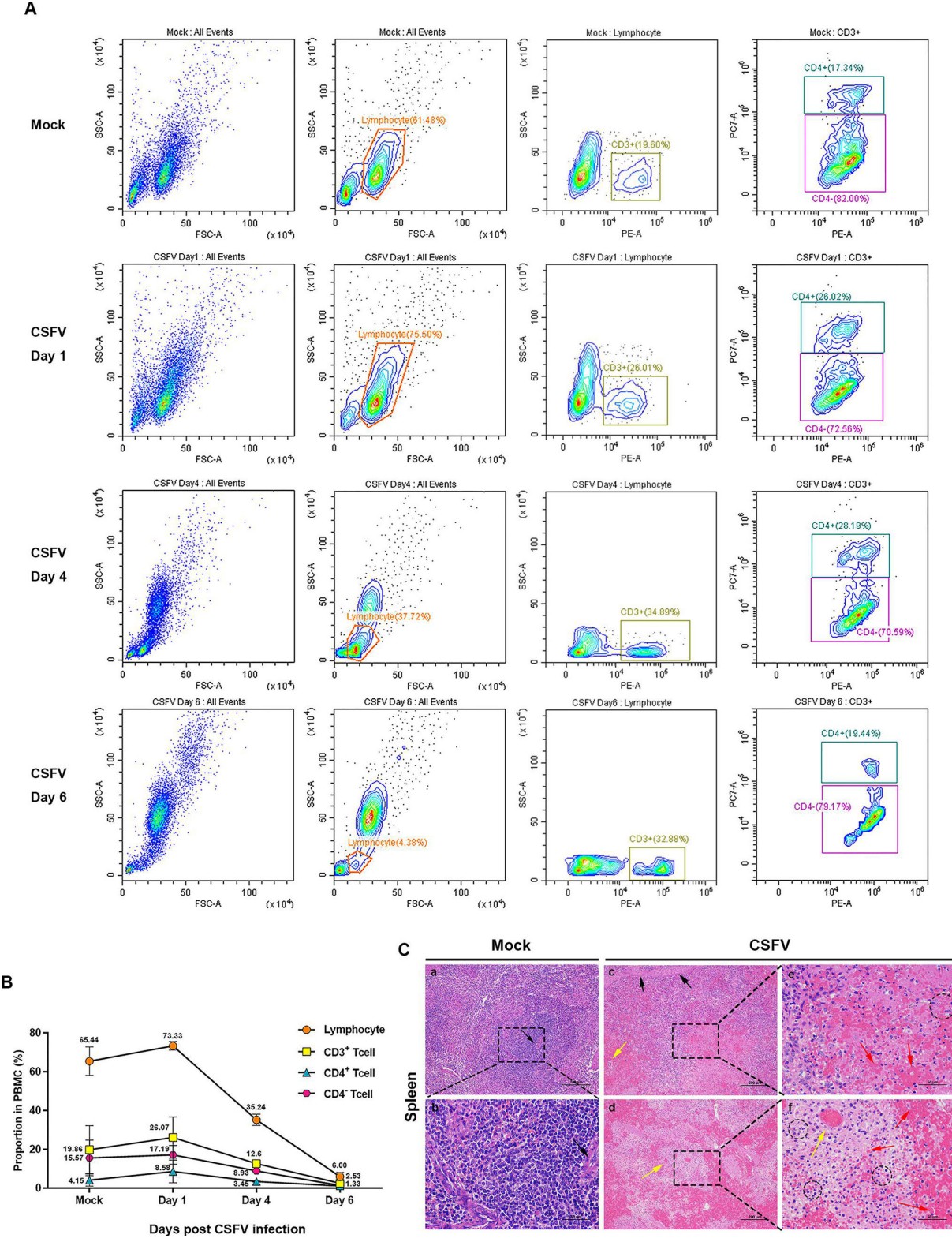

**FIG 1** CSFV-infected piglets cause peripheral blood T lymphocyte reduction and spleen necrosis. (A) Peripheral blood mononuclear cells were isolated from uninfected and CSFV Shimen strain-infected piglets at days 1, 4, and 6 to label the proportions of CD3$^+$ and CD4$^+$ T lymphocytes, respectively, by flow cytometry in the representative images. (B) Quantification of the ratio of CD3$^+$and CD4$^+$ T lymphocytes shown in panel A. (C) The experimental pigs were sacrificed at 7 dpi, and spleens were collected for tissue preparation and stained with hematoxylin and eosin, and then spleen's histopathological changes were analyzed with (Continued on next page)

**FIG 1** (Continued)

a NIKON Eclipse Ci biological microscope (Japan), under the magnification of 200× and 400×. Bar: 100 and 200 μm, respectively. Histopathological changes as indicated by differently colored arrows and dotted circles can be seen in the panel.

Lymphopenia is a major cause of immunosuppression. To explore the cause of lymphopenia in piglets with CSFV infection, we performed transcriptomic analysis of CD3$^+$ T lymphocytes in the peripheral blood as well as splenic lymphocytes (SPLs) of piglets infected with CSFV, respectively, and set up uninfected CD3$^+$ T lymphocytes as well as SPL controls. The DEGs were determined by the results of |log$_2$ fold change| >1 and $P_{adj}$ <0.05 by RNA sequencing. Compared with the control group, 293 genes were upregulated, and 161 genes were downregulated in the CD3$^+$ T lymphocyte-treated group (Fig. 2A); 831 genes were upregulated, and 962 genes were downregulated in the SPL-treated group (Fig. 2B). Significant differential genes related to cell death, including *MLKL*, an executive protein of programmed necrosis, were selected among the DEGs in both cell types; *IL1A*, a pleiotropic cytokine that accompanies the onset of inflammation; *CASP1*, a caspase involved in the execution phase of apoptosis. Cluster analysis showed that the CSFV-infected group had a good cluster-like relationship with the control group, indicating that the CSFV infection-induced gene transcriptome differences within CD3$^+$ T lymphocytes and spleen lymphocytes were small but significantly different between groups (Fig. 2C and D). The associated genes were subjected to pathway association analysis, which revealed a cellular network linking CSFV infection to lymphocyte death (Fig. 2E). Go functional annotation of DEGs showed that the most significantly enriched biological processes were defense response (Go: 0006952), and 66 DEGs were enriched. The most significant enrichment in molecular function was for chemokine receptor binding (Go: 0042379), with enrichment of 10 DEGs. The most significantly enriched gene in the cellular component was plasma membrane part (Go: 0044459), and 59 DEGs were enriched (Fig. 2F). The results of Kyoto Encyclopedia of Genes and Genomes (KEGG) analysis showed significant enrichment of cytokine-cytokine receptor interaction, Toll-like receptor signaling pathway, IL-17 signaling pathway, TNF signaling pathway, viral protein interaction with cytokines and cytokine receptors, and necroptosis pathway (Fig. 2G). We further selected a large number of enriched cell death-related genes to validate the accuracy of the transcriptome data, respectively (Fig. 2H and I), and the results showed that the quantitative reverse transcription polymerase chain reaction (qRT-PCR) was approximately the same as the RNA-seq results, indicating the reliability of the transcriptome data. All these data suggest that these pathways may play important roles in porcine peripheral blood CD3$^+$ T lymphocytes against CSFV infection and that CSFV infection-induced lymphopenia may be closely related to the significantly enriched necroptosis pathway.

## CSFV infection induces necroptosis in the spleen and PBMCs *in vivo*

To explore whether CSFV infection could cause necrosis in peripheral blood and spleen lymphocytes, by transmission electron microscopy, we detected an increased number of cells with distinct necrotic features, indicated by their enlarged cell volume, focal disruption of the cell membrane, plasma membrane leakage, and cell lysis (Fig. 3A and C), as well as a significantly increased number of autophagosomes (Fig. 3A and B), in PBMCs isolated from CSFV-infected piglets. Furthermore, excessive erythrocyte flooding (red arrows) was observed in the spleens of CSFV-infected piglets, consistent with (Fig. 1C). Similarly, both necrotic cells and autophagosome numbers were obviously increased (Fig. 3D through F), suggesting that there may be some association between CSFV infection-induced autophagy and necrotic cells *in vivo*. We further examined the protein levels of necroptosis markers, first determining the successful infection of PBMCs by CSFV through the expression of CSFV N$^{pro}$ protein and found that MLKL expression level, an executive protein of necroptosis, was significantly higher in PBMC cells from CSFV-infected piglets, along with inflammatory cytokines TNF-α and IL-1β were significantly expressed (Fig. 3G), indicating that the induction of inflammation by CSFV infection is

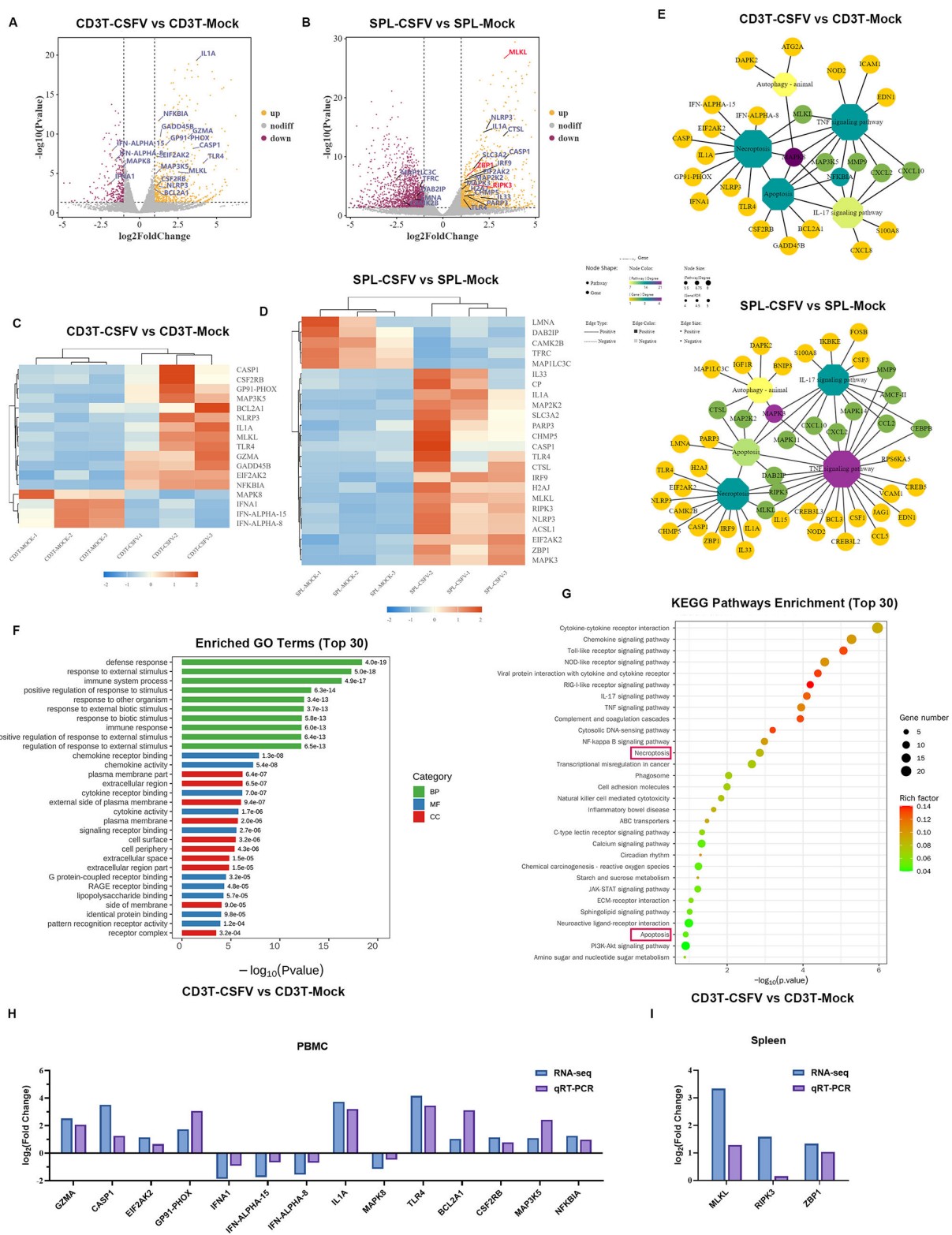

**FIG 2** Global omics analysis of CSFV-infected peripheral blood CD3$^+$ T lymphocytes and splenic lymphocytes. (A and B) Volcano plots of CSFV-infected peripheral blood CD3$^+$ T lymphocytes and spleen lymphocytes, respectively, to visually display the overall distribution of differential genes. (The DEGs with *P*-value <0.05, |log2FC| >1 were selected by R package software. Scatterplots represent individual genes; gray, genes with no significant difference; yellow, significantly differentially upregulated genes; purple, significantly differentially downregulated genes.) (C and D) Differential gene clustering plots of CSFV-infected peripheral blood CD3$^+$ T lymphocytes and spleen lymphocytes, respectively. According to the similarity of gene expression situation in each

**FIG 2** (Continued)

sample, the genes were clustered to visually demonstrate the difference of gene expression situation in different samples. (Each column represents a sample, and each row represents a gene. Red, upregulated genes; blue, downregulated genes.) (E) CSFV infects cellular networks associated with lymphocyte death. (Octagons represent pathways, circles represent genes, and network visualization generated by Cytoscape v.3.8.0.) (F) The Go functional annotation of DEGs in CSFV-infected peripheral blood CD3[+] T lymphocytes (green, biological process; blue, molecular function; red, cellular component). (G) The KEGG pathway enrichment analysis of DEGs from CSFV-infected peripheral blood CD3[+] T lymphocytes was performed based on KEGG pathways as background, the significance level of gene enrichment of each pathway was calculated by Fisher's exact test, and the top 30 enriched pathways were selected to draw a bubble plot. (H) PBMCs from CSFV-infected piglets at 7 dpi were isolated; RNA was extracted, and significantly enriched differential gene mRNA fold changes were quantified by qRT-PCR, and bars were plotted together with fold change from RNA-seq. (I) Spleens of CSFV-infected piglets at 7 dpi were collected; RNA was extracted, and significantly enriched differential gene mRNA fold changes were quantified by qRT-PCR, and bars were plotted together with fold change from RNA-seq.

likely accompanied by the onset of necroptosis. Furthermore, we found that CSFV infection significantly increased the mRNA levels of necroptosis hallmark genes *RIPK1*, *RIPK3*, and *MLKL* in PBMCs (Fig. 3I). In addition, mRNA levels of *TNF-α, TNFR1*, and *ZBP1*, upstream genes of necroptosis, were significantly increased in CSFV-infected PBMCs, whereas little effect on *Caspase-8* mRNA levels was observed (Fig. 3K). Meanwhile, CSFV infection was found to be able to promote the proportion of necrotic cells in PBMCs by flow cytometry analysis (Fig. 3H). Similarly, we detected significantly higher mRNA levels of *MLKL* and *ZBP1* in the spleen of CSFV-infected piglets, whereas no significant changes in mRNA levels of *RIPK1, RIPK3, and TNF-α* were observed (Fig. 3J and L). This may be attributed to significant differences among the RNA samples within the group. Furthermore, the high expression of necroptosis marker proteins RIPK1, RIPK3, and MLKL in the spleen of CSFV-infected piglets was validated by immunohistochemistry (IHC; Fig. 3M through O). The above results indicated that CSFV infection was able to induce PBMCs and spleen to undergo necroptosis *in vivo*. To explore whether CSFV-induced necroptosis has a tissue tropism, we examined the mRNA levels of necroptosis marker genes in different immune tissues, such as tonsil, lymph node, and thymus, and non-immune tissues, such as lung and kidney, respectively, and found that the expression of these genes was significantly promoted by CSFV in the thymus, significantly suppressed in kidney and tonsil, and unchanged in lung and lymph node (Fig. S1). The results suggest that the occurrence of necroptosis may be related to the viral load and the extent of the lesions in the infected tissue, with some tissue tropism, and also reflect to some extent that necroptosis as a viral defense mechanism may play an important role in combating CSFV infection.

## Necroptosis is induced at the early stage of CSFV infection but inhibited at the late stage *in vitro*

We found that CSFV infection-induced necroptosis has a certain tissue tropism, so what is the relationship between CSFV infection and necroptosis *in vitro*? Is this consistent *in vivo*? We cultured the isolated PBMCs *in vitro* and monitored the protein level and mRNA level changes of necroptosis marker genes *RIPK1*, *RIPK3*, *MLKL,* and *ZBP1* at 24, 48, and 60 h after CSFV infection; meanwhile, we determined the successful establishment of a PBMC model of CSFV infection *in vitro* by detecting the expression of CSFV N[pro] protein. Compared with control cells not infected with CSFV, CSFV infection for 24 h promoted the expression of necroptosis-related proteins, while it markedly inhibited the expression of RIPK1-, RIPK3-, MLKL-, and ZBP1-related proteins in PBMCs at 48 or 60 h (Fig. 4A and B). With longer infection time, CSFV could obviously increase *RIPK3* and *ZBP1* mRNA levels in PBMCs compared with the control, whereas it suppressed MLKL mRNA levels at 60 h of infection (Fig. 4G). This observation may appear contradictory to the results at the protein level but is consistent with the conclusion that CSFV infection can significantly induce necrotic apoptosis in PBMCs. As for the protein-level inhibition observed in the later stages, it is likely closely related to the autophagy process induced by CSFV. Furthermore, using an *in vitro* established cell model of CSFV infection of PK-15 and 3D4/21, similar to infected PBMC cells, we found that CSFV was able to promote the expression of necroptosis-related genes in PK-15 (Fig. 4C, D, and H) and 3D4/21 (Fig. 4E, F,

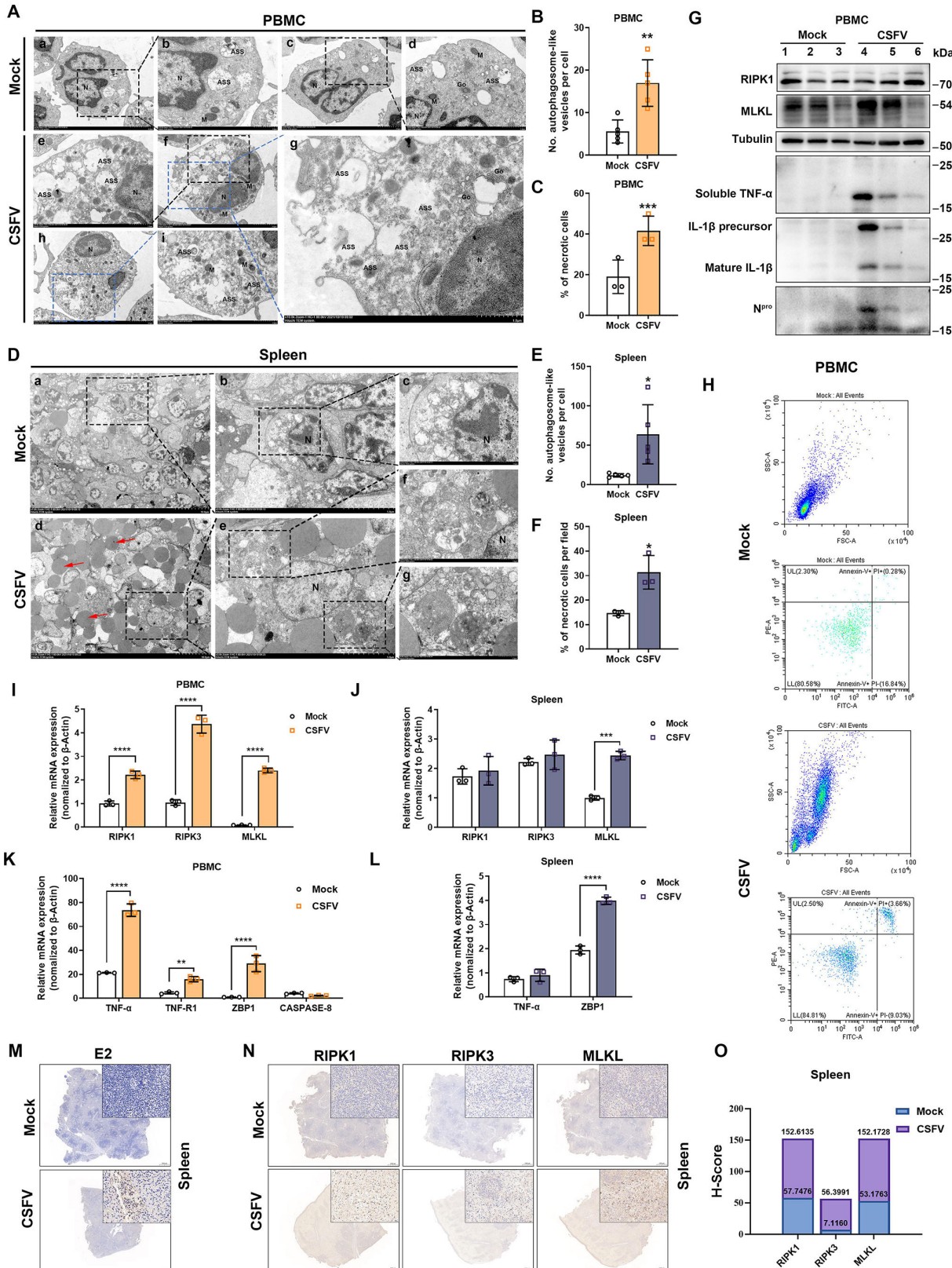

**FIG 3** Necroptosis is induced in PBMCs and spleens of CSFV-infected piglets *in vivo*. (A) PBMCs from CSFV-infected and uninfected piglets at 7 dpi were isolated, and necrotic cell features as well as autophagy-like vesicles were visualized by transmission electron microscopy (N, nucleus; M, mitochondria; Go, golgi; ASS, autophagosome). (B) Quantitative analysis of the number of autophagosomes observed in (A). Error bars indicate the mean (± SD) of five independent (Continued on next page)

**FIG 3** (Continued)

experiments. **, $P < 0.01$ (one-way ANOVA). (C) Quantitative analysis of the proportion of necrotic cells observed in (A). Error bars indicate the mean (± SD) of three independent experiments. ***, $P < 0.001$ (one-way ANOVA). (D) Spleens from CSFV-infected and uninfected piglets at 7 d were collected, and necrotic cell features as well as autophagy-like vesicles were visualized by transmission electron microscopy (N, nucleus; red arrow, red blood cell). (E) Quantitative analysis of the number of autophagosomes observed in (D). Error bars indicate the mean (± SD) of five independent experiments. *, $P < 0.05$ (one-way ANOVA). (F) Quantitative analysis of the proportion of necrotic cells observed in (D). Error bars indicate the mean (± SD) of three independent experiments. *, $P < 0.05$ (one-way ANOVA). (G) The protein levels of RIPK1, MLKL, TNF -α, IL-1β, and CSFV N$^{pro}$ in PBMC of CSFV-infected and non-infected piglets were detected by Western blot. (I and K) The mRNA expressions of *RIPK1*, *RIPK3*, *MLKL*, *TNF-α*, T*NF-R1*, *ZBP1,* and *Caspase-8* in PBMC of CSFV-infected and non-infected piglets were determined by qRT-PCR. The fold changes were related to the internal control β-actin. Error bars indicate the mean (± SD) of three independent experiments. **, $P < 0.01$; ***, $P < 0.001$; ****, $P < 0.0001$ (two-way ANOVA). (J and L) The mRNA expressions of *RIPK1*, *RIPK3*, *MLKL*, *TNF-α,* and *ZBP1* in spleens of CSFV-infected and non-infected piglets were determined by qRT-PCR. The fold changes were related to the internal control β-actin. Error bars indicate the mean (± SD) of three independent experiments. ***, $P < 0.001$ and ****, $P < 0.0001$ (two-way ANOVA). (H) Flow cytometry was used to determine the proportion of necrotic cells in PBMCs from CSFV-infected and uninfected piglets at 7 d. (M) The expression of CSFV E2 in the spleen of infected and uninfected piglets was detected by IHC. (N) The expression of RIPK1, RIPK3, and MLKL in the spleen of infected and uninfected piglets was detected by IHC. During this period, we fixed, embedded, and sectioned spleen tissues. The sections were then stained with specific antibodies against RIPK1, RIPK3, and MLKL, respectively. The appearance of similar contours in the sliced images is a result of the proximity of the tissue sections. (O) Immunohistochemical scoring of the detection indices in (N).

and I) cells at the preinfection stage and inhibit the expression of related genes at the later stage of infection as the infection time progressed. These data indicate that CSFV was able to induce necroptosis in the early phase of *in vitro* infection, whereas it was inhibited in the middle and late phases. The differential changes in necrotic apoptosis-related gene and protein levels induced by CSFV infection in PBMCs, PK-15, and 3D4/21 cells may be closely associated with the tissue tropism of CSFV. This may also be an important reason why CSFV infection *in vitro* fails to elicit cytopathic effects.

## CSFV NS4A interacts with RIPK3 and TRIM25 to inhibit necroptosis

In order to explore the molecular mechanism by which necroptosis is inhibited by CSFV infection *in vitro*, we overexpressed the major core protein (C) as well as the non-structural proteins P7, NS4A, NS4B, and NS5A of CSFV *in vitro* and observed significant colocalization of only GFP-NS4A and HA-RIPK3 in HEK-293T cells co-transfected with HA-RIPK3 and pGFP-C1, GFP-P7, GFP-C, GFP-NS4A, GFP-NS4B, GFP-NS5A, respectively (Fig. 5A), and RIPK interacted with CSFV NS4A protein was further verified by co-immunoprecipitation (Co-IP) assay; additionally, the protein level of RIPK3 was obviously decreased in the presence of NS4A (Fig. 5C). Meanwhile, we transfected PK-15 cells with plasmids expressing different viral proteins and found that RIPK3 protein levels were decreased, and MLKL protein was also decreased in the presence of CSFV NS4A alone compared with transfection with empty vector pEGFP-C1 (Fig. 5B). Previous studies in our laboratory identified TRIM25, an E3 ubiquitin ligase, as a potential interacting molecule with CSFV NS4A protein by mass spectrometry, and we also further demonstrated the relevant functional relationship by Co-IP (Fig. 5D and 2A) and laser confocal assay (Fig. S2B). Notably, overexpression of TRIM25 promoted the protein expression of NS4A (Fig. 5D). Moreover, NS4A specifically mediated the decrease in RIPK3, which was more pronounced when NS4A levels were gradually increased (Fig. S2C). Interestingly, as NS4A levels increased, so did TRIM25 expression (Fig. S2C). It was previously reported that TRIM25 was able to interact with RIPK3 and mediate its degradation, acting as a brake molecule for necroptosis (44), then is this phenomenon also present in PK-15 cells? We found that RIPK3 was gradually degraded as the expression level of TRIM25 increased (Fig. S2D), and there was a significant spatial colocalization of CSFV NS4A with TRIM25, RIPK3 (Fig. 5E). Furthermore, significant co-localization of NS4A with TRIM25 (Fig. S3A) and NS4A with RIPK3 (Fig. S3B) was observed during CSFV infection. Taking a step further, we transfected the CSFV NS4A-expressing plasmid into PK-15 cells overexpressing TRIM25 and knocking down TRIM25, respectively, to explore whether NS4A specifically mediates the degradation of RIPK3 via TRIM25. The results showed that the overexpression of TRIM25 specifically promoted the degradation of RIPK3 by NS4A, whereas RIPK1 levels were not affected by TRIM25 (Fig. 5F). Correspondingly, the

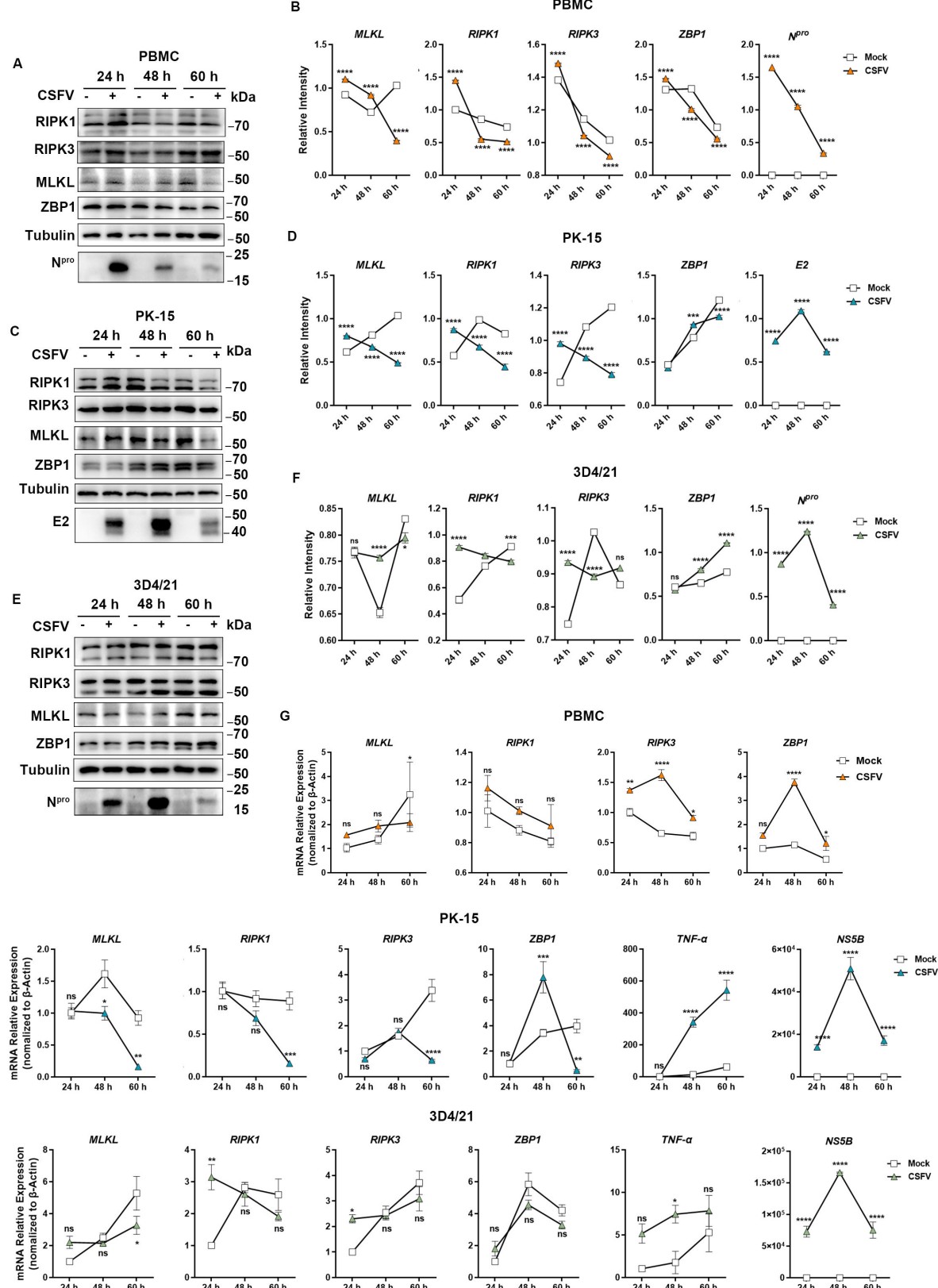

**FIG 4** Necroptosis was induced at the early stage and inhibited at the late stage of CSFV infection *in vitro*. To establish a PBMC cell model of CSFV infection *in vitro*, cells were collected at 24, 48, and 60 h post-infection, and protein expression and mRNA levels of necroptosis markers including RIPK1, RIPK3, MLKL, and

**FIG 4** (Continued)

ZBP1 were detected by Western blot (A) and qRT-PCR (G). (B) Gray-scale value analysis of the detection indexes in (A); the relative intensity was related to the internal control Tubulin. Error bars indicate the mean (± SD) of three independent experiments. ns, $P \geq 0.05^*$; $P < 0.05$; **, $P < 0.01$; and ***, $P < 0.001$ (two-way ANOVA). To establish a PK-15 cell model of CSFV infection *in vitro*, cells were collected at 24, 48, and 60 h post-infection, and protein expression and mRNA levels of necroptosis marker genes including RIPK1, RIPK3, MLKL, and ZBP1 were detected by Western blot (C) and qRT-PCR (H). (D) Gray-scale value analysis of the detection indexes in (C); the relative intensity was related to the internal control Tubulin. Error bars indicate the mean (± SD) of three independent experiments. ns, $P \geq 0.05^*$; $P < 0.05$; **, $P < 0.01$; and ***, $P < 0.001$ (two-way ANOVA). To establish a 3D4/21 cell model of CSFV infection *in vitro*, cells were collected at 24, 48, and 60 h post-infection, and protein expression and mRNA levels of necroptosis markers including RIPK1, RIPK3, MLKL, and ZBP1 were detected by Western blot (E) and qRT-PCR (I). (F) Gray-scale value analysis of the detection indexes in (E); the relative intensity was related to the internal control Tubulin. Error bars indicate the mean (± SD) of three independent experiments. ns, $P \geq 0.05^*$; $P < 0.05$; **, $P < 0.01$; and ***, $P < 0.001$ (two-way ANOVA).

knockdown of TRIM25 was able to specifically restore the inhibitory effect of NS4A on RIPK3, and neither NS4A nor TRIM25 had an obvious regulatory effect on RIPK1 (Fig. S2E). Moreover, it was not difficult to find that CSFV NS4A modulated the expression level of the necroptosis executioner MLKL via TRIM25 (Fig. 5F and 2E).

## CSFV infection inhibits necroptosis via autophagy/mitophagy *in vitro*

Previously, we found that CSFV was able to induce necroptosis at the initial stage of infection, which was inhibited at the later stage. Previous studies have found that after CSFV infects host cells, the replication of viruses can be promoted by activating the autophagy/mitophagy pathway in cells, and the utilization of autophagic vesicles provides a favorable site for the replication of viruses (40, 42). Necroptosis, as an effective defense mechanism of the organism, is activated at the initial stage of CSFV infection. So, during the interaction between CSFV and the host organism, is it possible for CSFV to interfere with or even inhibit necroptosis through the autophagy pathway, thus achieving persistent infection within the cells?

PK-15 (Fig. 6A) cells uninfected and infected with CSFV for 48 h were, respectively, treated with autophagy agonist rapamycin. The results showed that rapamycin elevated the autophagic flux in PK-15 cells and promoted the expression of CSFV Npro protein. More importantly, rapamycin further strengthened the inhibitory effect of CSFV infection on necroptosis-related proteins, especially RIPK3 and MLKL. On this basis, we further measured the expression levels of mitophagy-related proteins and key molecules of necroptosis by treating CSFV-uninfected and -infected PK-15 (Fig. 6B) cells with mitophagy inducer carbonyl cyanide 3-chlorophenylhydrazone (CCCP). The results showed that CCCP increased the mitophagy level in PK-15 cells and further strengthened the inhibitory effect of CSFV infection on necroptosis key molecules, especially RIPK3 and MLKL. In addition, we further verified this phenomenon using flow cytometry by pretreating CSFV-infected PK-15 cells with necroptosis inducer (TSZ; TNF-α, Smac mimetic, and z-VAD-Fmk) autophagy agonist rapamycin (Fig. 6C). Furthermore, we verified that RIPK3 expression was significantly elevated in the presence of CSFV infection and inhibited the autophagy pathway in CSFV-infected cells by introducing inhibitors at different stages of autophagy or transfecting with shRNA targeting interfering autophagy-related genes (Fig. 6D). These results suggest that CSFV most likely inhibits necroptosis by activating the autophagy pathway.

## Necroptosis restricted CSFV infection, while induction of autophagy by CSFV antagonizes necroptosis for persistent infection

Exploring the role of necroptosis in CSFV infection, we found that the reduction in CSFV infection as determined by released infectious virus titers was observed in PK-15 and 3D4/21 cells overexpressing RIPK3 and MLKL (Fig. 6E). Correspondingly, we treated CSFV-infected PK-15 cells with TSZ, the relative expression of CSFV *NS5B* markedly decreased in TSZ-treated PK-15 cells (Fig. 6F), and the protein expression of CSFV Npro obviously decreased with increasing doses of TSZ treatment (Fig. 6G), suggesting that necroptosis may significantly reduce CSFV infection. To further confirm the limiting effect of necroptosis on CSFV infection, we treated CSFV-infected PK-15 cells with necrosulfonamide

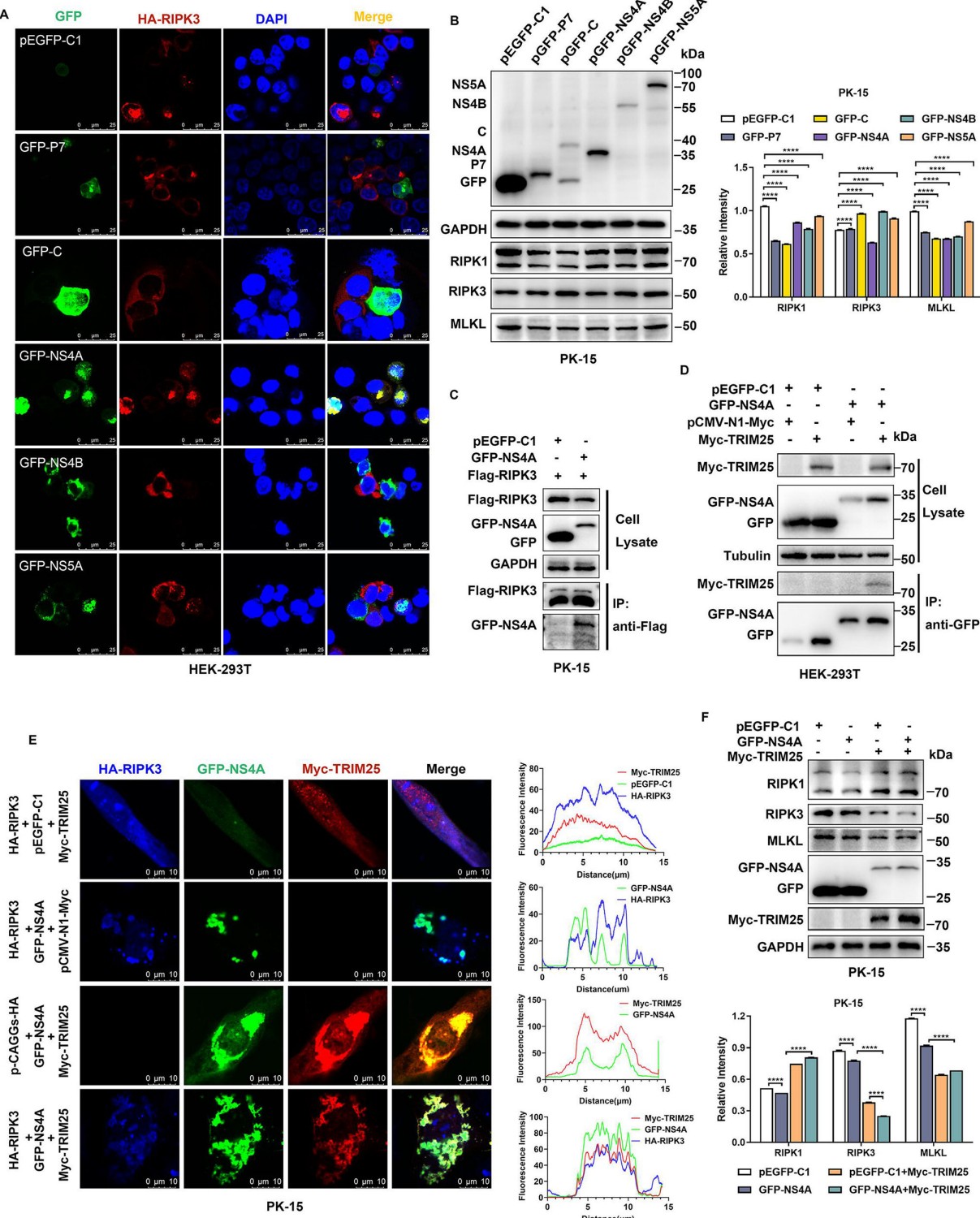

**FIG 5** CSFV NS4A interacts with RIPK3 and TRIM25 to inhibit necroptosis. (A) HEK-293T cells were co-transfected with HA-RIPK3 together with pGFP-C1, GFP-P7, GFP-C, GFP-NS4A, GFP-NS4B, or GFP-NS5A for 24 h, fixed cells were incubated with anti-HA tag primary antibody and then stained with Alexa fluor 555-conjugated anti-Rabbit-IgG secondary antibody (red), and nuclei were stained with DAPI. Scale bars: 25 µm. (B) pGFP-C1, GFP-P7, GFP-C, GFP-NS4A, GFP-NS4B, and GFP-NS5A were transfected into PK-15 cells for 24 h, and the cells were lysed. The expression levels of viral proteins, RIPK1, RIPK3, and MLKL, were detected by Western blot. The right panel was gray-scale value analysis, the relative intensity was related to the internal control GAPDH. Error bars indicate the mean (± SD) of three independent experiments. ****, *P* < 0.0001 (two-way ANOVA). (C) FLAG-RIPK3 was co-transfected with pEGFP-C1 and GFP-NS4A, (Continued on next page)

**FIG 5** (Continued)

respectively, into PK-15 cells, and the cells were lysed; Co-IP assay and Western blot analysis were performed. (D) Myc-TRIM25 was co-transfected with pEGFP-C1 and GFP-NS4A in HEK-293T cells, while a control group of cells was set up in which pCMV-N1-Myc was co-transfected with pEGFP-C1 and GFP-NS4A, cells were lysed; Co-IP experiments and Western blot analysis were performed. (E) PK-15 cells were co-transfected with three constructs, HA-RIPK3, Myc-TRIM25 with GFP-NS4A, and a control construct of cells co-transfected with HA-RIPK3, Myc-TRIM25 with pEGFP-C1, HA-RIPK3, pCMV-N1-Myc with GFP-NS4A, Myc-TRIM25, pCAGGs-HA with GFP-NS4A, The fixed cells were incubated with anti-HA primary antibody and anti-Myc primary antibody, respectively, and then stained with Alexa fluor 555-conjugated anti-Rabbit-IgG Mouse antibody (red) and Dylight 405-labeled anti-Rabbit-IgG secondary antibody (blue) (left panel). Scale bars: 10 µm. The fluorescence intensity profiles of HA-RIPK3 (blue), GFP-NS4A(green), and Myc-TRIM25(red) were analyzed using Image J plugin and plotted by GraphPad Prism 9 (right panel). (F) Myc-TRIM25 was co-transfected with pEGFP-C1 and GFP-NS4A in PK-15 cells, while a control group of cells was set up in which pCMV-N1-Myc was co-transfected with pEGFP-C1 and GFP-NS4A; the cell lysates were detected by Western blot. The gray-scale value analysis was related to the internal control GAPDH (lower panel). Error bars indicate the mean (± SD) of three independent experiments. ****, $P < 0.0001$ (two-way ANOVA).

(NSA), an inhibitor of MLKL, and we could obviously observe that the inhibition of MLKL expression by NSA obviously promoted CSFV N$^{pro}$ protein expression (Fig. 6H). In PK-15 and 3D4/21 cells treated with both TSZ and rapamycin, the expression of CSFV N$^{pro}$ drastically decreased with the addition of TSZ but was obviously restored with further rapamycin treatment (Fig. 6I). Taken together, these data strongly indicated that necroptosis is a limiting host factor for CSFV infection and that CSFV infection-induced autophagy was able to subvert this host defense mechanism to promote its sustained replication.

## CSFV NS4A synergizes TRIM25 to induce mitophagy

Previously, we showed that CSFV NS4A can promote the expression of TRIM25, which is likely to act as a key molecule linking CSFV NS4A and RIPK3. To gain insight into the biological function of TRIM25, we overexpressed TRIM25 in PK-15 and 3D4/21 cells, respectively, and measured the levels of p-mTOR, LC3-II, and P62. Overexpression of TRIM25 significantly increased LC3-II/LC3-I conversion and inhibited the expression of autophagy substrate P62 compared with the control, while the expression level of p-mTOR was not affected by TRIM25 (Fig. 7A). We further examined the effect of TRIM25 on the expression levels of mitochondria-associated proteins HSP60, TOMM20, COX IV, and VDAC. The results showed that the expression of mitochondria-associated proteins TOMM20, COX IV, and VDAC was significantly decreased in PK-15 cells overexpressing TRIM25 compared with the control, as was the expression of HSP60 and COX IV in 3D4/21 cells (Fig. 7B). To obtain a clearer and specific view of the cellular localization of TRIM25 to mitochondria and the effect of TRIM25 on mitochondrial number and morphology, we immunostained mitochondria with the mitochondrial probe Mito-Tracker, which allows us to observe that TRIM25 can partially localize to mitochondria and that PK-15 overexpressing TRIM25 appears granular to the mitochondria of 3D4/21 cells, and mitochondria of cells transfected with empty vector appear filamentous (Fig. 7C). The effect of TRIM25 on the formation and number of autophagosome-like vesicles was further examined by transmission electron microscopy. It was found that clear mitochondrial cristae (black arrows) could be observed mostly in the mitochondria of cells transfected with empty vector, while the mitochondrial cristae structure of PK-15 cells overexpressing TRIM25 was disrupted, and some mitochondria were wrapped by double- or single-membrane structure, that is, typical mitophagosome structure appeared (red arrows), indicating that TRIM25 can perturb the mitochondrial metabolic activity of cells and lead to abnormal mitochondrial morphology (Fig. 7D). In addition, the number of mitochondria encapsulated by autophagosome-like vesicles was significantly higher in PK-15 cells overexpressing TRIM25 than in empty vector-transfected cells. It is suggested that the TRIM25 protein can lead to the abnormality of cellular mitochondrial morphology and induce the occurrence of mitophagy.

Consistent with this previous report, CSFV NS4A was also able to partially colocalize with mitochondria; however, the knockdown of TRIM25 was able to suppress the expression of CSFV NS4A and inhibit its degree of colocalization with mitochondria (Fig.

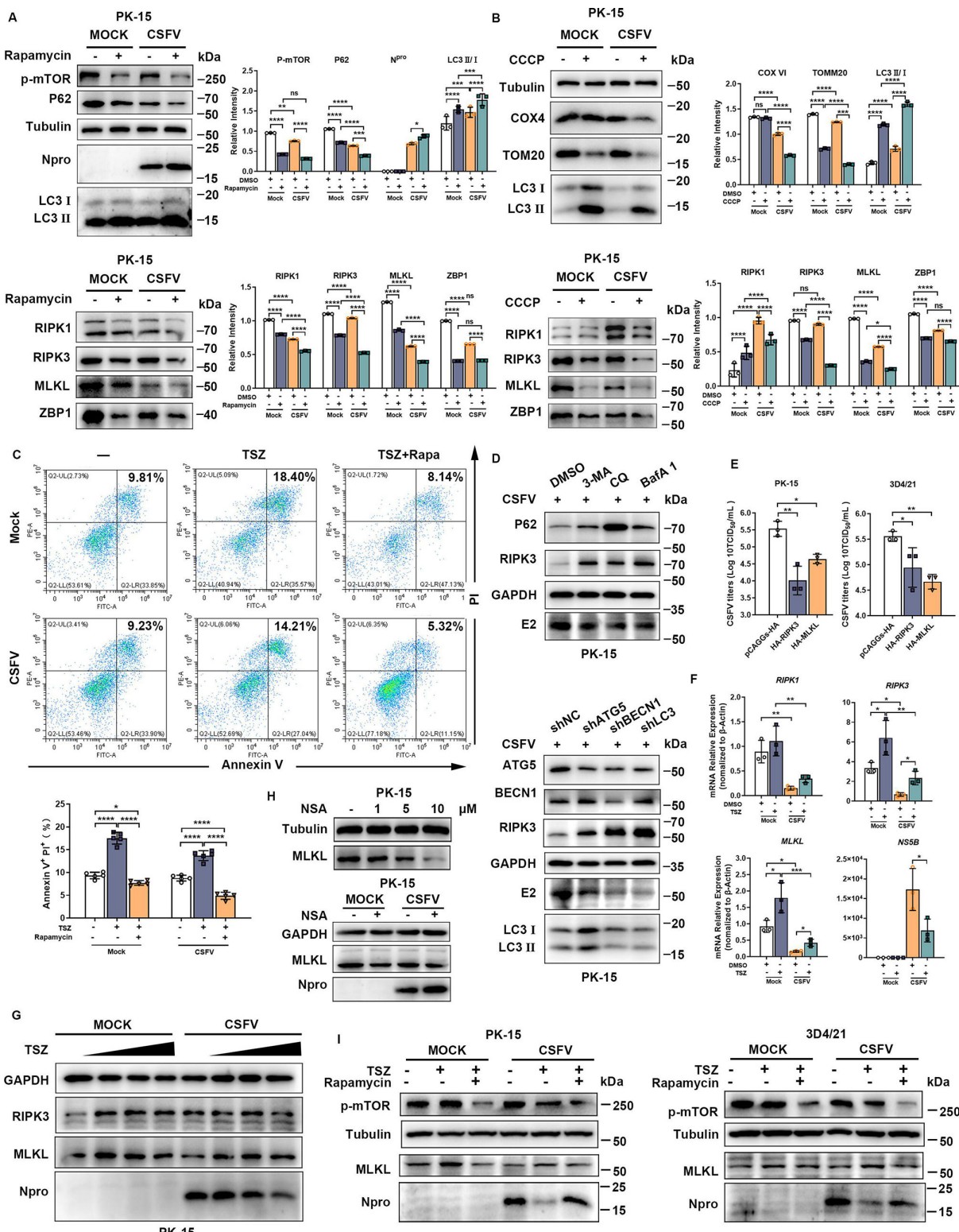

**FIG 6** CSFV inhibits necroptosis through autophagy/mitophagy and necroptosis-restricted CSFV infection *in vitro*. (A) PK-15 cells were infected with CSFV (MOI = 1) for 48 h, rapamycin was treated for 6 h, and an uninfected cell control was set up. Cell lysates were then detected by Western blot (left panel). Relative intensity related to internal control Tubulin was analyzed using Image J plugin and plotted by GraphPad Prism 9 (right panel). Error bars indicate the mean (± SD) of three independent experiments. ns, $P \geq 0.05$; *, $P < 0.1$; **, $P < 0.01$; ***, $P < 0.001$; ****, $P < 0.0001$ (two-way ANOVA). (B) PK-15 cells were infected with CSFV (MOI = 1) for 48 h, CCCP treated for 6 h, and an uninfected cell control was set up. Cell lysates were then detected by Western blot (left panel). Relative intensity

**FIG 6** (Continued)

related to internal control Tubulin was analyzed using Image J plugin and plotted by GraphPad Prism 9 (right panel). Error bars indicate the mean (± SD) of three independent experiments. ns, $P \geq 0.05$; *, $P < 0.1$; ***, $P < 0.001$; ****, $P < 0.0001$ (two-way ANOVA). (C) PK-15 cells were infected with CSFV (MOI = 1) for 48 h and treated with TSZ (20 ng/mL TNF-α, 100 nM Smac mimetic, and 20 µM z-VAD-Fmk) alone or TSZ together with rapamycin for 6 h, and a control of uninfected cells was set up. Necrotic cells were identified by flow cytometry with PI and Annexin V staining. A representative plot of the data is shown in the left panel, and the mean (± SD) of three independent experiments is shown in the right panel. *, $P < 0.1$ and ****, $P < 0.0001$ (two-way ANOVA). (D) PK-15 cells infected with CSFV (MOI = 1) were transfected with Flag-RIPK3 for 24 h and then treated with 3-methyladenine (3-MA, 5 mM), chloroquine (CQ, 20 µM), or bafilomycin A1 (Baf A1, 100 nM) for 6 h, and cell lysates were detected by Western blot (upper panel). Flag-RIPK3 was co-transfected with shNC, shATG5, shBECN1, and shLC3 into CSFV-infected PK-15 cells, and cell lysates were detected by Western blot (lower panel). (E) PK-15 (left panel) and 3D4/21 (right panel) cells infected with CSFV (MOI = 1) were transfected with HA-RIPK3 or MLKL for 48 h, and pCAGGs-HA was transfected with a control group. The cell supernatants were collected for TCID50 assay. Error bars indicate the mean (± SD) of three independent experiments. *, $P < 0.1$ and **, $P < 0.01$ (one-way ANOVA). (F) PK-15 cells infected with CSFV (MOI = 1) were treated with TSZ for 6 h, and control of uninfected cells was set up. The mRNA expressions of *RIPK1*, *RIPK3*, *MLKL*, and CSFV *NS5B* in the cell lysates were determined by qRT-PCR. The fold changes were related to the internal control β-actin. Error bars indicate the mean (± SD) of three independent experiments. *, $P < 0.1$; **, $P < 0.01$; ***, $P < 0.001$ (one-way ANOVA). (G) PK-15 cells infected with CSFV (MOI = 1) were treated with increasing amounts of TSZ (wedge shaped, 0; 20 ng/mL TNF-α, 100 nM Smac mimetic, and 10 µM z-VAD-Fmk; 40 ng/mL TNF-α, 100 nM Smac mimetic, and 10 µM z-VAD-Fmk; 40 ng/mL TNF-α, 100 nM Smac mimetic, and 20 µM z-VAD-Fmk) for 6 h, and the cell lysates were detected by Western blot. (H) PK-15 cells were treated with increasing amounts of necrosulfonamide (NSA; wedge shaped, 0, 1, 5, and 10 µM), and the cell lysates were detected by Western blot (upper panel). PK-15 cells infected with CSFV (MOI = 1) were treated with NSA (5 µM), and dimethyl sulfoxide (DMSO) was treated with a control group. The cell lysates were detected by Western blot (lower panel). (I) PK-15 (left panel) and 3D4/21 (right panel) cells were infected with CSFV (MOI = 1) for 48 h and treated with TSZ (20 ng/mL TNF-α, 100 nM Smac mimetic, and 20 µM z-VAD -Fmk) alone or TSZ together with rapamycin for 6 h, and a control of uninfected cells was set up. The cell lysates were detected by Western blot.

S4D). CSFV NS4A, as an important non-structural protein of CSFV, forms a replication-associated complex with NS3, NS4B, NS5A, and NS5B to participate in viral replication. At present, little is known about the role of CSFV NS4A protein in the process of mitophagy. We found that overexpression of CSFV NS4A promoted the degradation of autophagy substrate P62 as well as the conversion of LC3-II/LC3-I, and more importantly, its ability to inhibit the expression of mitochondria-related proteins (Fig. 7E), and a higher number of mitochondria encapsulated by autophagosome-like vesicles in NS4A overexpressing PK-15 cells than in empty vector-transfected cells were observed by transmission electron microscopy (Fig. S4A). Even more, overexpression of TRIM25 was able to deepen the effect of NS4A on the degradation of mitochondria-related proteins in PK-15 (Fig. 7E and F) and 3D4/21 (Fig. S4B) cells and the induction of autophagy. Conversely, the knockdown of TRIM25 was able to specifically restore the inhibitory effect of NS4A on mitochondria-related proteins in PK-15 (Fig. 7G and H) and 3D4/21 (Fig. S4C) cells and attenuate the induction effect on autophagy. We co-transfected recombinant plasmid HA-NS4A expressing CSFV NS4A with Myc-TRIM25 and Si TRIM25, respectively, into PK-15, to monitor autophagic flux using mRFP-GFP-LC3, and strong aggregation of LC3 with orange fluorescence was observed in NS4A overexpressing, especially PK-15 overexpressing both NS4A and TRIM 25, whereas it was reduced upon siRNA-mediated knockdown of TRIM 25 (Fig. 7I). We further employed the mitophagy dual fluorescent reporter plasmid Mito-mRFP-EGFP to observe the delivery of mitochondria-lysosomes in above co-transfected HA-NS4A and Myc-TRIM25/Si TRIM25 cells. The results showed that the mitochondria of PK-15 (Fig. 7J and S4E) and 3D4/21 (Fig. S4F) cells overexpressing both NS4A and TRIM25 exhibited orange-yellow fluorescence, whereas the mitochondria of cells transfected with empty vector exhibited yellow fluorescence, and the mitochondria of cells transfected with both NS4A and Si TRIM25 exhibited greenish yellow fluorescence. The progress of autophagy/mitophagy was judged according to the quenching degree of green fluorescence, indicating that CSFV NS4A cooperated with TRIM25 to induce autophagy, causing the degradation of mitochondria by cell lysosomes.

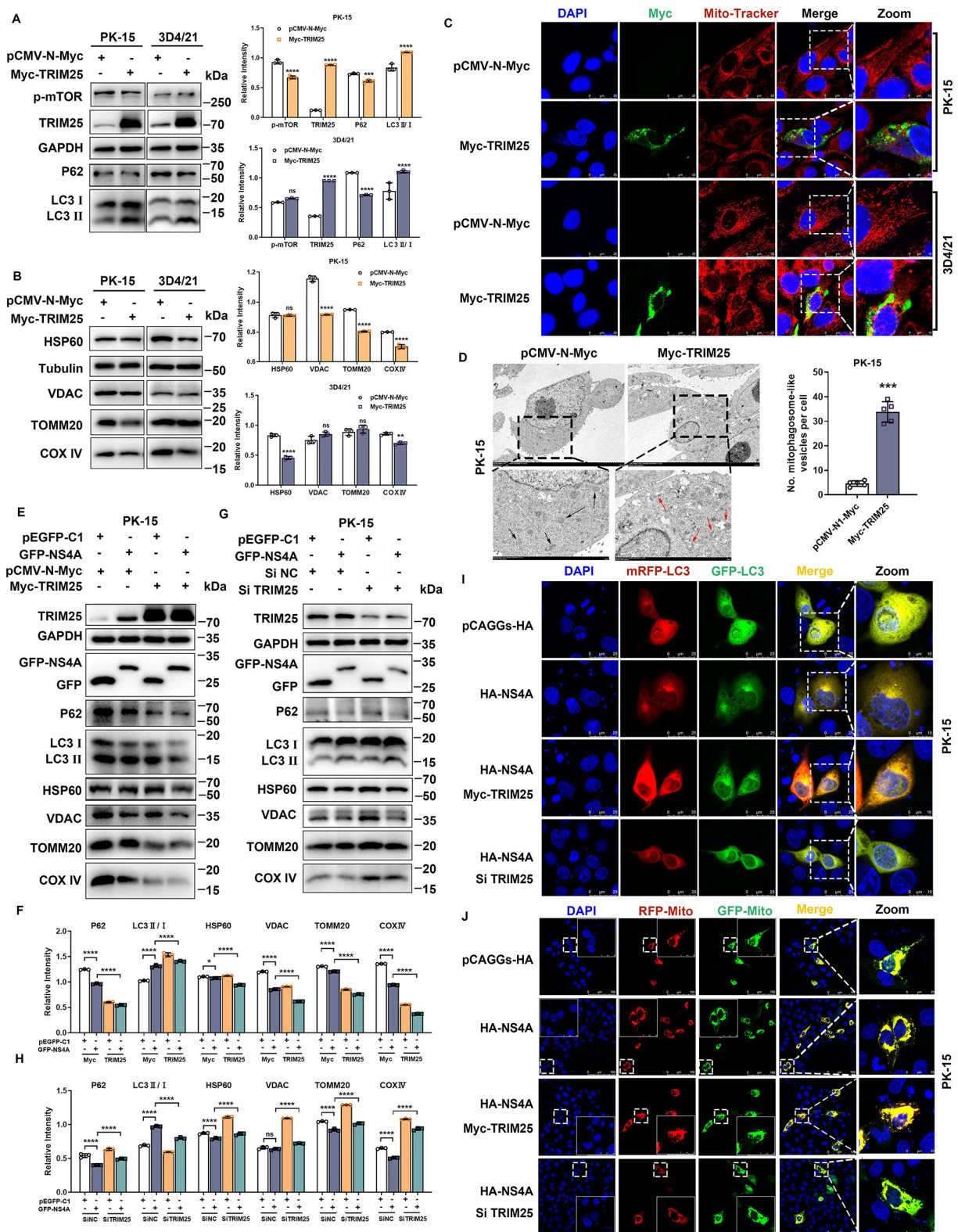

**FIG 7** CSFV NS4A synergizes TRIM25 to induce mitophagy. (A and B) PK-15 and 3D4/21 cells were transfected with Myc-TRIM25 for 24 h, and pCMV-N1-Myc cells were transfected with control group. Expression levels of autophagy-associated proteins p-mTOR, LC3-II, and P62 and mitochondrial-associated proteins HSP60, TOMM20, COX IV, and VDAC in cell lysis products were detected by Western blot (left panel). Relative intensity related to internal parameters was analyzed using Image J plugin and plotted by GraphPad Prism 9 (right panel). Error bars indicate the mean (± SD) of three independent experiments. ns, $P \geq 0.05$; **, $P <$

**FIG 7 (Continued)**

0.01; \*\*\*, $P < 0.001$ (two-way ANOVA). (C) PK-15 and 3D4/21 cells were transfected with Myc-TRIM25 for 24 h, and pCMV-N1-Myc cells were set up as a control group. The cells were immunostained with Mito-Tracker, incubated with anti-Myc primary antibody, followed by Alexa fluor 488-conjugated anti-Rabbit-IgG secondary antibody (green), and the nuclei were stained with DAPI. Scale bars: 25 and 10 μm. (D) Mitochondrial structures and autophagy-like vesicles were visualized by transmission electron microscopy (left panel), and the number of autophagic vesicles with encapsulated mitochondria was quantified (right panel). Error bars indicate the mean (± SD) of five independent experiments. \*\*\*, $P < 0.0005$ (one-way ANOVA). (E) Myc-TRIM25 was co-transfected with pEGFP-C1 and GFP-NS4A in PK-15 cells, respectively, while a control group of cells was set up in which pCMV-N1-Myc was co-transfected with pEGFP-C1 and GFP-NS4A, respectively. Expression levels of autophagy-associated proteins p-mTOR, LC3-II, and P62 and mitochondrial-associated proteins HSP60, TOMM20, COX IV, and VDAC in cell lysis products were detected by Western blot. (F) The relative intensity of the detection indexes in (E), related to the internal control GAPDH, was analyzed with Image J plugin and plotted by GraphPad Prism 9. Error bars indicate the mean (± SD) of three independent experiments. \*, $P < 0.1$; and \*\*\*\*, $P < 0.0001$ (two-way ANOVA). (G) Si TRIM25 was co-transfected with pEGFP-C1 and GFP-NS4A in PK-15 cells, respectively, while a control group of cells was set up in which Si NC was co-transfected with pEGFP-C1 and GFP-NS4A, respectively. Expression levels of autophagy-associated proteins p-mTOR, LC3-II, and P62 and mitochondrial-associated proteins HSP60, TOMM20, COX IV, and VDAC in cell lysis products were detected by Western blot. (H) The relative intensity of the detection indexes in (G), related to the internal control GAPDH, was analyzed with Image J plugin and plotted by GraphPad Prism 9. Error bars indicate the mean (± SD) of three independent experiments. ns, $P \geq 0.05$ and \*\*\*\*, $P < 0.0001$ (two-way ANOVA). (I) HA-NS4A was co-transfected with Myc-TRIM25 and Si TRIM25 into PK-15 cells, respectively, and a cell control transfected with HA-NS4A only and pCAGGs-HA only was set up, along with dual-fluorescent reporter plasmid mRFP-GFP-LC3 to visualize the autophagic flux. (J) HA-NS4A was co-transfected with Myc-TRIM25 and Si TRIM25 into PK-15 cells, respectively, and a cell control transfected with HA-NS4A only and pCAGGs-HA only was set up, along with mitophagy dual-fluorescent reporter plasmid Mito-mRFP-EGFP to visualize the mitochondria-lysosome delivery. The fluorescence intensity of GFP-Mito (green) and RFP-Mito (red) was analyzed in Fig. S4E with Image J plugin and plotted by GraphPad Prism 9. Error bars indicate the mean (± SD) of at least three independent experiments. \*, $P < 0.05$; \*\*\*, $P < 0.001$; \*\*\*\*, $P < 0.0001$ (two-way ANOVA).

## RIPK3 localizes to the outer mitochondrial membrane and is degraded by TRIM25-induced mitophagy under CSFV infection

Previously, we found that CSFV NS4A specifically mediates the degradation of RIPK3 via TRIM25, the specific molecular mechanism of which remains unclear. Currently, there are mainly two main intracellular protein degradation systems, the ubiquitin-proteasome system (UPS) and autophagy-lysosome pathway (47), we further explored which degradation system TRIM25 mainly employs to regulate the degradation of RIPK3 after CSFV infection. To this end, CSFV-infected PK-15 cells were co-transfected with plasmids expressing RIPK3 and TRIM25, and then the cells were treated with the protease inhibitor MG132, rapamycin, and the autophagy inhibitor bafilomycin A1 (Baf A1), respectively. The results showed that both MG132 and Baf A1 could increase the protein expression level of RIPK3 (Fig. 8A). Consistent with previous reports, we found that TRIM25 was able to ubiquitinate RIPK3 after CSFV infection to mediate its degradation via the UPS (Fig. 8B). Co-transfection of HA-RIPK3 and HA-MLKL separately with GFP-LC3 into PK-15 cells allowed to observe that RIPK3, but not MLKL, could co-localize with LC3 (Fig. 8C and 5A). Furthermore, mitochondria in RIPK3-overexpressed PK-15 cells were immunostained with Mito-tracker, a mitochondrial probe, to observe the spatial distribution of RIPK3 with mitochondria. Interestingly, RIPK3 was able to wrap around the outer mitochondrial membrane, and the degree of co-localization of RIPK3 with mitochondria was obviously enhanced in the knockdown of TRIM25 (Fig. 8D). In contrast, there was no significant colocalization of MLKL with mitochondria in either the presence or absence of Si TRIM25 (Fig. S5B). In addition, we labeled the lysosomes of PK-15 cells co-expressing RIPK3 and LC3 with the lysosomal outer membrane protein CD63 under CSFV infection conditions and could find that RIPK3 was able to co-localize with LC3 to lysosomes, and the intensity of co-localization with LC3 was reduced after knockdown of TRIM25 (Fig. 8E).

## CSFV promotes the expression of TRIM25, which enhances CSFV replication

TRIM25, as a key molecule linking CSFV NS4A and TRIM25, how is its interaction with CSFV? We examined the mRNA levels of *TRIM25* in different tissues including lung, kidney, and lymph nodes in CSFV-uninfected and -infected piglets and found that CSFV promotes *TRIM25* transcription in different tissues, with distinct tissue tropism for *TRIM25* (Fig. 8F). Western blot results showed that CSFV *in vitro* infection of PK-15 and 3D4/2 cells for 24 and 48 h was able to obviously promote the protein expression of TRIM25 (Fig. 8G).

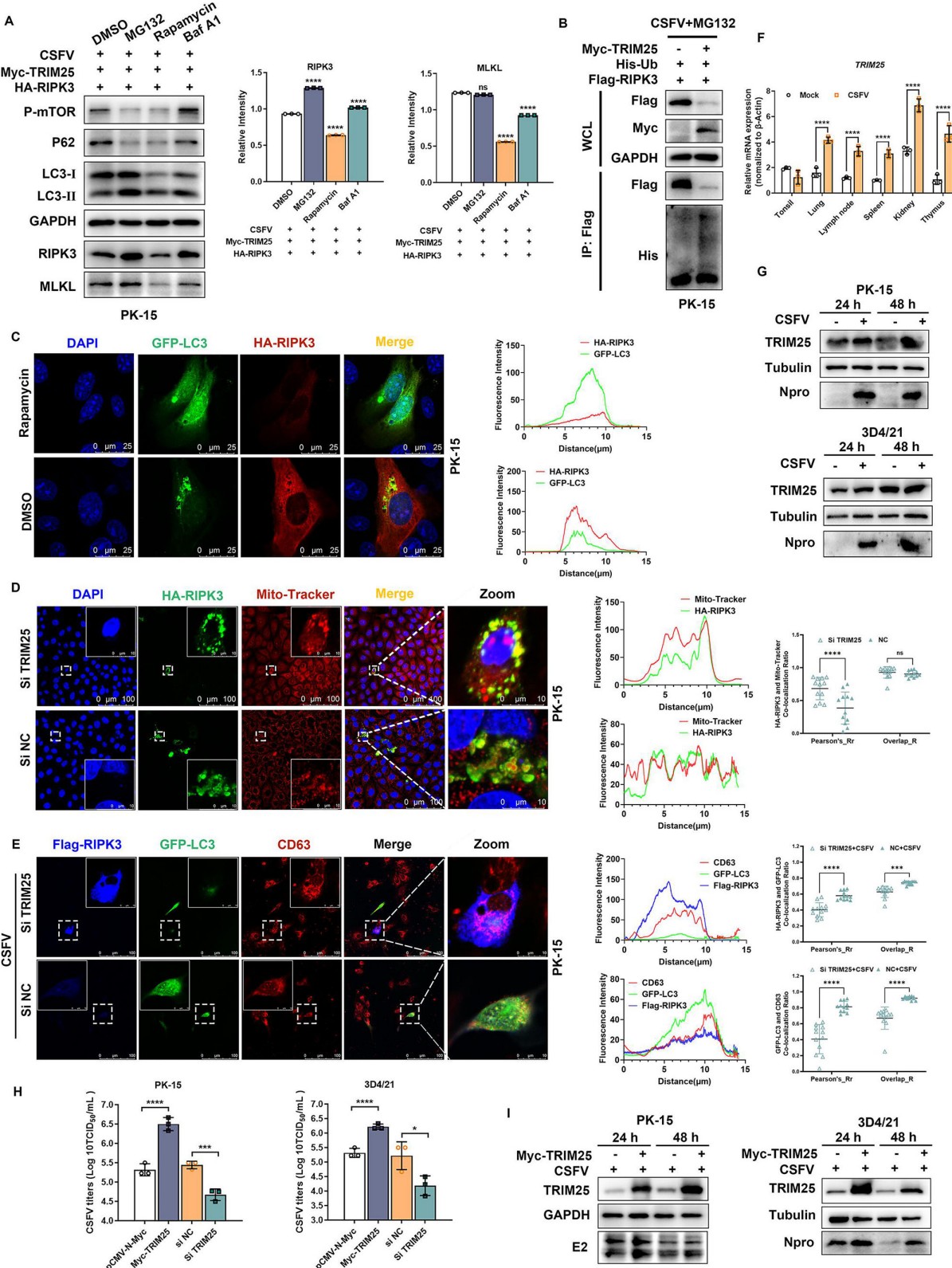

**FIG 8** RIPK3 is able to be degraded by TRIM25-induced mitophagy under CSFV infection. (A) PK-15 cells infected with CSFV (MOI = 1) were co-transfected with Myc-TRIM25 and HA-RIPK3 for 24 h and then treated with MG132 (10 mM), rapamycin (100 nM), or Baf A1 (100 nM) for 6 h, and cell lysates were detected by Western blot (left panel). Relative intensity related to internal control GAPDH was analyzed using Image J plugin and plotted by GraphPad Prism 9 (right panel). Error bars indicate the mean (± SD) of three independent experiments. ns, $P \geq 0.05$ and ****, $P < 0.0001$ (one-way ANOVA). (B) PK-15 cells infected with CSFV (Continued on next page)

**FIG 8** (Continued)

(MOI = 1) were co-transfected with Myc-TRIM25, Flag-RIPK3, and His-Ub for 24 h and then treated with MG132 (10 mM) for 6 h, and the cells were lysed and used for IP experiments. (C) HA-RIPK3 and GFP-LC3 were co-transfected into PK-15 cells for 24 h and then treated with rapamycin (100 nM) for 6 h while setting up a control group of cells treated with dimethyl sulfoxide (DMSO). The cells were incubated with anti-HA primary antibody, followed by Alexa fluor 555-conjugated anti-Rabbit-IgG secondary antibody (red), and the nuclei were stained with DAPI (left panel). Scale bars: 100 and 25 μm. The fluorescence intensity profiles of GFP-LC3 (green) and HA-RIPK3 (red) were analyzed using Image J plugin and plotted by GraphPad Prism 9 (right panel). (D) PK-15 cells were co-transfected with HA-RIPK3 together with Si TRIM25 for 24 h while setting up a control group of co-transfected Si NC cells, the cells were immunostained with Mito-Tracker, incubated with anti-HA primary antibody, followed by Alexa fluor 488-conjugated anti-Mouse-IgG secondary antibody (green), and nuclei were stained with DAPI (left panel). Scale bars: 100 and 10 μm. The fluorescence intensity profiles of HA-RIPK3 (green) and Mito-Tracker (red) were analyzed using Image J plugin and plotted by GraphPad Prism 9 (middle panel). Pearson's correlation and overlap co-efficient of HA-RIPK3 and Mito-Tracker were analyzed using Image J plugin and plotted by GraphPad Prism 9 (right panel). Error bars indicate the mean (± SD) of at least three independent experiments. ns, $P \geq 0.05$ and *, $P < 0.1$ (one-way ANOVA). (E) PK-15 cells infected with CSFV (MOI = 1) were co-transfected with Si TRIM25, Flag-RIPK3, and GFP-LC3 for 24 h while setting up a control group of co-transfected Si NC cells, the fixed cells were incubated with anti-Flag primary antibody and anti-CD63 primary antibody, respectively, and then stained with Alexa fluor 555-conjugated anti-Mouse-IgG secondary antibody (red) and Dylight 405-labeled anti-Rabbit-IgG secondary antibody (blue) (left panel). Scale bars: 10 μm. The fluorescence intensity profiles of Flag-RIPK3 (blue), GFP-LC3(green), and CD63(red) were analyzed using Image J plugin and plotted by GraphPad Prism 9 (middle panel). Pearson's correlation and overlap co-efficient were analyzed using Image J plugin and plotted by GraphPad Prism 9 (right panel). Error bars indicate the mean (± SD) of at least three independent experiments. ***, $P < 0.001$ and ****, $P < 0.0001$ (one-way ANOVA). (F) The mRNA expressions of *TRIM25* in the tonsil, lung, lymph node, spleen, kidney, and thymus of CSFV-infected and non-infected piglets were determined by qRT-PCR. The fold changes were related to the internal control β-actin. Error bars indicate the mean (± SD) of three independent experiments. ****, $P < 0.0001$ (two-way ANOVA). (G) To establish PK-15 (upper panel) and 3D4/21 (lower panel) cell models of CSFV infection *in vitro*, cells were collected at 24 and 48 h post-infection, and protein expression levels of TRIM25 and CSFV $N^{pro}$ were detected by Western blot. (H) PK-15 (left panel) and 3D4/21 (right panel) cells infected with CSFV (MOI = 1) were transfected with Myc-TRIM25 or Si TRIM25 for 48 h, and pCMV-N1-Myc or Si NC cells were transfected with a control group. The cell supernatants were collected for $TCID_{50}$ assay. Error bars indicate the mean (± SD) of three independent experiments. *, $P < 0.1$; ***, $P < 0.001$; ****, $P < 0.0001$ (one-way ANOVA). (I) PK-15 (left panel) and 3D4/21 (right panel) cells infected with CSFV (MOI = 1) were transfected with Myc-TRIM25 for 24 and 48 h, and pCMV-N1-Myc cells were transfected with a control group. Expression levels of TRIM25 and CSFV E2/$N^{pro}$ in cell lysis products were detected by Western blot.

To further explore the role of TRIM25 in CSFV infection, a TRIM25 expression plasmid was transfected into PK-15 and 3D4/21 cells, respectively, followed by CSFV infection. Immunoblotting showed that the expression of CSFV E2 protein was significantly increased in PK-15 cells overexpressing TRIM25 compared with the control, and the expression of CSFV $N^{pro}$ protein was also significantly increased in 3D4/21 cells overexpressing TRIM25 compared with the control (Fig. 8I). We knocked down *TRIM25* expression using specific interfering RNA targeting or transfected *TRIM25* expression plasmids into PK-15 and 3D4/21 cells, followed by detection of viral titers in the cell supernatant after 48 h of CSFV infection. The results showed that functional silencing of *TRIM25* significantly suppressed CSFV titers in cell supernatants, whereas overexpression of *TRIM25* promoted increased CSFV infectivity titers (Fig. 8H).

## CSFV promotes RIPK3 autophagic degradation through the binding of autophagy receptor NDP52

Selective autophagy requires the engagement of autophagy receptors, which specifically recognize and traffic autophagy substrates, thereby targeting the degradation of autophagy substrates, endowing the material dependent on autophagic degradation under extremely sophisticated dynamic regulation (22, 25). We sought to identify the cargo receptor involved in the autophagic degradation of RIPK3, and the Co-IP results showed that RIPK3 was able to interact with P62 and NDP52, and overexpression of NDP52 was able to obviously inhibit the expression of RIPK3 compared with other cargo receptors (Fig. 9A). Consistent with this result, we were able to observe a clear co-localization of NDP52 with RIPK3 in the PK-15 cytoplasm (Fig. 9B). Notably, the distribution of RIPK3 protein expressed alone was more diffuse in cells, whereas in cells simultaneously expressing both proteins, the distribution of RIPK3 was more clustered and highly coincident with NDP52 (Fig. 9B). In addition, Baf A1 treatment of cells was able to more prominently observe the colocalization of RIPK3, LC3, and NDP52 in CSFV-infected PK-15 cells (Fig. 9C). Furthermore, significant co-localization of NS4A with NDP52 (Fig. S3B) was

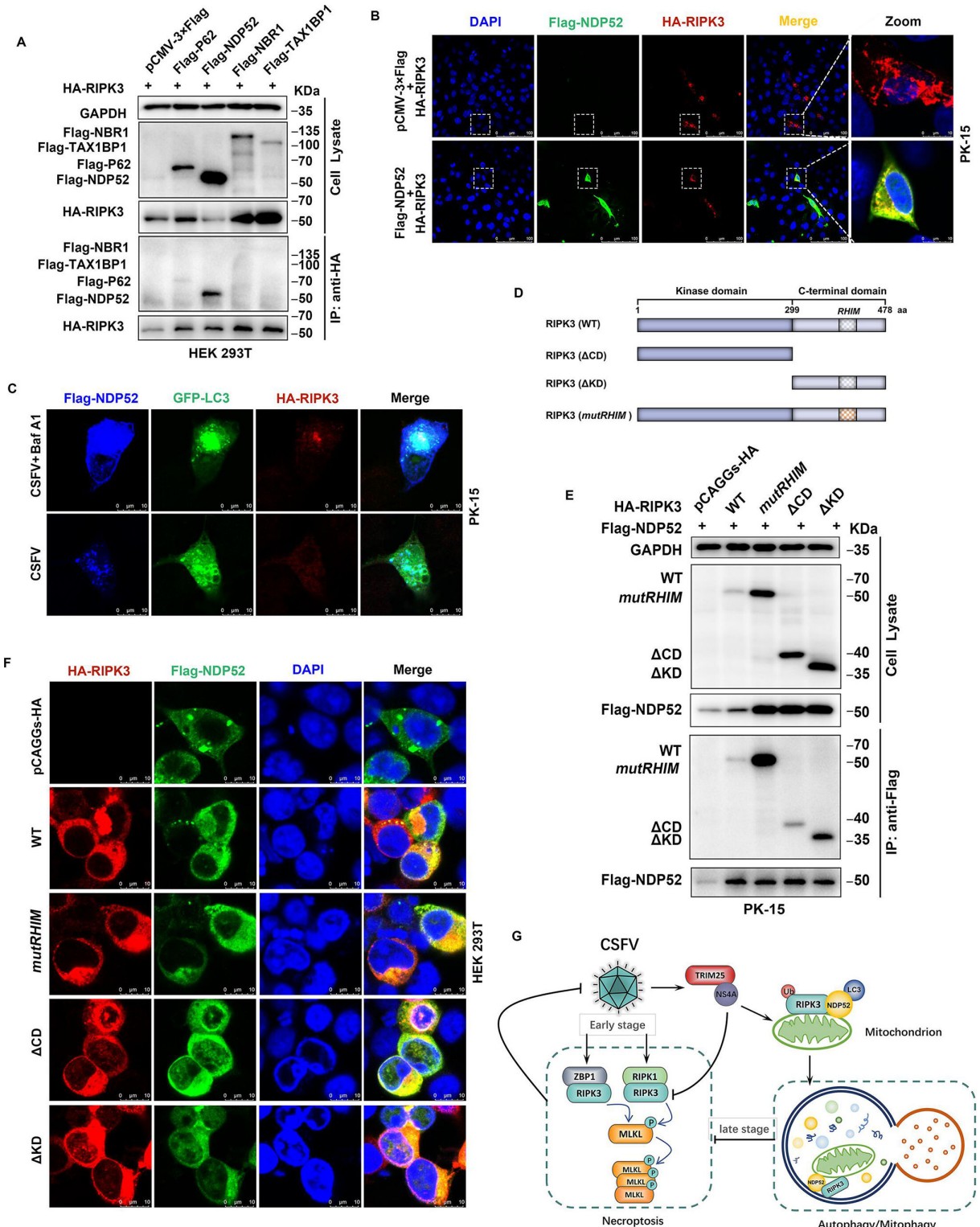

**FIG 9** The autophagy receptor NDP52 binds to RIPK3 and promotes its autophagic degradation. (A) HEK-293T cells were co-transfected with HA-RIPK3 together with pCMV-3× Flag, Flag-P62, Flag-NDP52, Flag-NBR1, or Flag-TAX1BP1 for 24 h, the cells were lysed and used for Co-IP experiments, and cell lysates were detected by Western blot. (B) Flag-NDP52 and HA-RIPK3 were co-transfected into PK-15 cells for 24 h while setting up a control group of cells transfected with pCMV-3× Flag. The fixed cells were incubated with anti-Flag primary antibody and anti-HA primary antibody, followed by Alexa fluor 488-conjugated anti-Mouse-IgG secondary antibody (green) and Alexa fluor 555-conjugated anti-Rabbit-IgG secondary antibody (red), and the nuclei were stained with DAPI. Scale bars: 100 and 10 µm. (C) PK-15 cells infected with CSFV (MOI = 1) were co-transfected with Flag-NDP52, GFP-LC3, and HA-RIPK3 for 24 h and then treated

**FIG 9** (Continued)

with Baf A1 (100 nM) for 6 h while setting up a control group of cells treated with dimethyl sulfoxide, the fixed cells were incubated with anti-Flag primary antibody and anti-HA primary antibody, respectively, and then stained with Alexa fluor 555-conjugated anti-Rabbit-IgG secondary antibody (red) and Dylight 405-labeled anti-Mouse-IgG secondary antibody (blue). Scale bars: 10 µm. (D) Schematic of RIPK3 domain structure and deletion/mutation structure. (E) PK-15 cells were co-transfected with Flag-NDP52 together with pCAGGs-HA, Flag-RIPK3 WT, Flag-RIPK3 *mutRHIM*, Flag-RIPK3 ΔCD, or Flag-RIPK3 ΔKD for 24 h, the cells were lysed and used for Co-IP experiments, and cell lysates were detected by Western blot. (F) HEK 293T cells were co-transfected with Flag-NDP52 together with pCAGGs-HA, Flag-RIPK3 WT, Flag-RIPK3 *mutRHIM*, Flag-RIPK3 ΔCD, or Flag-RIPK3 ΔKD for 24 h, the fixed cells were incubated with anti-Flag primary antibody and anti-HA primary antibody, followed by Alexa fluor 488-conjugated anti-Mouse-IgG secondary antibody (green) and Alexa fluor 555-conjugated anti-Rabbit-IgG secondary antibody (red), and the nuclei were stained with DAPI. Scale bars: 100 and 10 µm. (G) Proposed a model by which CSFV infection *in vitro* limits necroptosis via the autophagy pathway. *In vitro*, the induction of autophagy by CSFV at a later stage of infection clearly restricts necroptosis. Mechanistic studies revealed that CSFV NS4A protein promoted TRIM25 expression, synergistically induced the occurrence of mitophagy, targeted the autophagic degradation of RIPK3 to block the progression of necroptosis occurrence, and achieved persistent viral infection. Interestingly, we found that RIPK3 was able to specifically localize at the outer mitochondrial membrane, and the autophagy receptor NDP52 was most likely involved in the autophagic degradation of RIPK3 during CSFV infection.

observed during CSFV infection. We speculate that NDP52 may play an important role in regulating the autophagic degradation of RIPK3 in CSFV-infected cells.

To determine which domain of NDP52 interacts with RIPK3, we constructed constructs for truncation of the C-terminal domain (ΔCD, aa 1–299) and kinase domain (ΔKD, aa 300–478) of a truncation plasmid of RIPK3. In addition to mutating the RHIM (aa 453–470), the *mutRHIM* of RIPK3 was generated by overlap extension PCR, changing "VQIG" from 460 to 463 to "AAAA" (Fig. 9D). The results showed that NDP52 was able to interact with all three mutant or truncated forms of RIPK3 (Fig. 9E and F). Autophagy receptor proteins, on one hand, were able to recognize and bind autophagy substrates through a ubiquitin-associated (UBA) domain-dependent or -independent manner while anchored on the autophagosome membrane through their LIRs, and subsequently, the autophagosome and lysosome fuse, and the substrates are degraded in the lysosome (26, 48, 49). NDP52, as a UBA-containing receptor, can bind ubiquitinated autophagy substrates, and the ability of NDP52 to interact with different mutations or truncations of RIPK3 may be related to its presence of multiple ubiquitination sites across domains.

## DISCUSSION

The occurrence and development of infectious diseases are essentially a process of continuous mutual games between pathogenic microorganisms and their hosts. Viruses, as intracellular hosts, long-term "struggle" with the host during the evolution process, leading to the emergence of a highly complex and elaborate anti-infective immune system and mechanisms in the body; meanwhile, the virus also acquires various mechanisms to escape the body's immune response, which can lead to a worse response.

CSFV has a strong tropism for epithelial cells and immune cells and is a typical immunosuppressive virus that seriously damages the hematopoietic and immune systems. Viremia can occur early in CSFV infection, followed by immunopathological features of lymphopenia including B lymphocytes, cytotoxic T cells, and helper T cells, and lymphopenia is the leading cause of immunosuppression. Previously, it was shown that high titers of CSFV can be detected in bone marrow at an early stage of viral infection, and peripheral T lymphopenia is closely related to the apoptotic damage of bone marrow lymphocytes caused by CSFV (50). Further studies found that granulocytopenia and disordered bone marrow function were the results of hematopoietic cell apoptosis due to the interplay of viral and host mechanisms (51). Autophagy is generally considered as a cell survival mechanism and is closely related to apoptosis (52, 53). During CSFV *in vivo* infection, autophagy leads to the death of tissue cells in the immune organs of infected pigs and mediates the reduction of T-zone lymphocytes (54). However, a previous study found that instead of clearing CSFV from infected cells, the autophagy pathway was utilized by CSFV to provide favorable conditions for its replication in cells (55). The pathogenesis features resulting from CSFV infection are similar to those caused by a "cytokine storm," i.e., the disorder of massive secretion of

inflammation-related cytokines is closely related to disease progression (56). Necroptosis distinguishes itself from apoptosis by its pro-inflammatory role, necroptosis with rupture of the cell membrane, accompanied by content release, and in particular, the inflammatory factors released initiate a cascade of inflammatory responses. Earlier studies have long considered apoptosis as the main mechanism underlying the lymphopenia syndrome upon CSFV infection. However, an increased number of cells with distinct necrotic features in PBMCs and spleens of peripheral immune organs of CSFV-infected piglets were observed in the present study, which were accompanied by the inflammatory-related factors TNF-α and IL-1β, the expression of necroptosis executioner MLKL was elevated in PBMC cells from CSFV infected piglets, and IHC also further validated the high expression of necroptosis marker proteins RIPK1, RIPK3, and MLKL in the spleen of CSFV-infected piglets. Showed that CSFV infection induced necroptosis occurrence in PBMCs and spleen *in vivo*.

Cell death is an essential component of the host's immune response against invading microbial pathogens. However, it is not surprising for the fact that some viruses can balance host defenses against infection (57, 58). Many viruses with non-cytopathic effects, including CSFV, can hijack various mechanisms to relieve ongoing interference from host cells to remain infected. In our further *in vitro* assays, it was not difficult to find that CSFV infection of PK-15 and 3D4/21 cells could induce necroptosis at the early stage, while it inhibited at the middle and late stages, which might also be an important reason why CSFV infection *in vitro* could not cause cytopathic effects. A similar phenomenon occurs in the context of murine cytomegalovirus infection, a virus that inhibits necroptosis of murine cells by disrupting RHIM-dependent signal transduction (15). In addition, human cytomegalovirus has also been shown to prevent necrotic cell death in human fibroblasts and monocytes (28, 59). α-Members of the herpesvirus subfamily, HSV-1 and herpes simplex virus type 2, exhibit opposing activities in manipulating necroptosis, depending on the host species (60). CSFV induces necroptosis in PBMCs and spleen *in vivo* but inhibits necroptosis levels in porcine kidney-derived cell line (PK-15) and alveolar macrophage cell line (3D4/21) *in vitro* at 60 dpi. On one hand, the induction of necroptosis is closely related to the levels of autophagy induced during CSFV infection, suggesting that autophagy induction may precede necroptosis, ultimately leading to the suppression of necroptosis levels. On the other hand, CSFV infection displays different tissue tropisms and induces varying degrees of pathology in different tissues. For example, it can cause splenic marginal hemorrhage and infarction as well as dense subcapsular petechiae in the kidneys. However, CSFV infection *in vitro* does not induce cellular pathology. This suggests that the activation/inhibition of necroptosis by CSFV *in vivo* and *in vitro* is of great relevance to the corresponding tissue/target cell types.

The process of programmed necrosis is accompanied by autophagy, and programmed necrosis can activate autophagy, whereas its induced activation of the necrosome complex can in turn occur downstream of autophagosome formation (61). This illustrates that there is a complex link between programmed necrosis and autophagy that may simultaneously occur during cell death (28). CSFV was able to reach the peak of viral replication at 48 h of infection *in vitro*, accompanied by the induction of a strong autophagic response. Rather paradoxically, the necroptosis process at this point is progressively inhibited with a continued extension of the CSFV infection time, which is known to be long, and it is reasonable to speculate that there is some corollary link between these two modes of cell death in CSFV infection conditions. As expected, we treated CSFV-infected cells with an autophagy/mitophagy agonist, and unsurprisingly, the level of necroptosis induced at the initial stage of CSFV infection was markedly suppressed. The observation that RIPK3, but not MLKL, was able to localize to the outer mitochondrial membrane and specifically co-localizes with LC3 further explains the outcome of necroptosis inhibition by treatment with mitophagy agonists. Importantly, we found that both necroptosis inducer TSZ treatment and overexpression of RIPK3 and MLKL significantly inhibited CSFV replication and infection progression, whereas the

necroptosis executioner MLKL clearly promoted CSFV replication when inhibited by NSA, suggesting that necroptosis is a limiting host factor for CSFV infection and that CSFV infection-induced autophagy disintegrates this host defense mechanism to promote its sustained replication.

To further explore the molecular mechanism by which CSFV infection inhibits necroptosis *in vitro*, we found that CSFV NS4A protein was able to interact with the E3 ubiquitin ligase TRIM25, which was first identified ubiquitously because of its antiviral role, and TRIM25-mediated ubiquitination activation of RIG-I is an important step in initiating intracellular antiviral responses (62). In addition, TRIM25 can regulate host antiviral capacity through other pathways, including negative regulation of RIG-I, activation of melanoma differentiation-associated gene 5, enhancing the activity of zinc-finger antiviral proteins (ZAP), and inhibiting viral RNA synthesis (63, 64). Interestingly, we found in our study that CSFV NS4A protein obviously promoted the expression of trim25 in a dose-dependent manner, while TRIM25 did not play an antiviral role as it did in other virus infections, unexpectedly obviously promoted the infection progress of CSFV, correspondingly, TRIM25 was also able to stabilize the expression of NS4A protein. Our further study revealed that NS4A was able to form a complex with TRIM25, RIPK3, following a previous report that TRIM25 was a novel "brake" molecule targeting the necroptotic key protein RIPK3 and was able to modify RIPK3 by ubiquitination to inhibit TNF-induced cell death. We verified that CSFV NS4A could suppress the expression of RIPK3 via TRIM25 functional silencing, and more importantly, CSFV NS4A induced mitochondrial fission and mitophagy with the assistance of TRIM25, suggesting that TRIM25 is likely to serve as a specific and critical target for CSFV infection escape. Based on our finding that TRIM25 mediates RIPK3 degradation through both the UPS and autophagy-lysosome pathways under CSFV infection conditions, a biological phenomenon somewhat contradicted by the finding that TRIM25 mainly degrades RIPK3 through the ubiquitination pathway, we speculated that CSFV engagement might alter the RIPK3 degradation pathway to some extent. In addition, it has been mentioned in other reports that RIPK3 was able to be ubiquitinated and then undergo lysosome-dependent degradation (65, 66), consistent with our finding of enhanced colocalization of RIPK3 with lysosomes after CSFV infection.

Autophagic degradation of substrates requires selective recognition and trafficking of cargo receptors, and we screened for evidence that autophagic degradation of RIPK3 may require, but is not limited to, the involvement of NDP52, but we did not find a specific site where NDP52 interacts with RIPK3. As in our study, RIPK3 with truncated C-terminal domain, kinase domain, or mutated RHIM region was still able to interact with NDP52, we hypothesized that NDP52 acts as an autophagy receptor containing a UBA and depends on ubiquitin-binding substrates for ubiquitination, whereas porcine RIPK3, similar to human RIPK3, contains multiple ubiquitination sites spanning domains. In subsequent studies, we could sequentially construct RIPK3 variants with serial ubiquitination site mutations, further confirming the binding site of NDP52. In addition, the association of TRIM25 with NDP52 is still missing in this study, and it remains to be confirmed that TRIM25 mediates autophagic degradation of RIPK3 via the NDP52 receptor under CSFV infection, and the relationship axis remains to be refined.

Collectively, we found that CSFV infection *in vivo* was able to activate necroptosis in PBMCs and spleen, leading to lymphocyte necrosis. Whereas, CSFV infection markedly inhibited necroptosis by inducing autophagy *in vitro*. It was further found that the CSFV NS4A protein interacts with the E3 ubiquitin ligase TRIM25 and is targeted to mediate the autophagic degradation of RIPK3, thereby blocking the progression of necroptosis to achieve persistent viral infection (Fig. 9G). Our findings shed further light on the mechanisms that cause T lymphocyte exhaustion during CSFV infection, provide evidence supporting an important role for CSFV in counteracting host cell necrosis, and enrich our knowledge of pathogens that evade this host defense.

## MATERIALS AND METHODS

### Cell and virus

The swine kidney cell line PK-15 (ATCC, CCL-33) and HEK293T (ATCC, CRL-1573) cells were grown in DMEM (Thermo Fisher Scientific, 11995500) supplemented with 10% (vol/vol) fetal bovine serum (FBS, Thermo Fisher Scientific, 10099), and macrophage cell line 3D4/21 (ATCC, CRL-2845) cells were maintained in complete RPMI 1640 medium (Thermo Fisher Scientific, 1175093) containing 10% FBS. Cells were cultured at 37°C in a 5% $CO_2$ incubator. The CSFV-Shimen strain was stocked in our laboratory and proliferated in the cultured PK-15 cells. Virus titers were determined with 50% tissue culture infective dose ($TCID_{50}$) assays.

### Animal experiment

All procedures were conducted following the regulations of the Laboratory Animal Ethics Committee of South China Agricultural University. Ten 2-month-old Tibetan pigs negative for the porcine reproductive and respiratory syndrome, pseudorabies virus, and porcine parvovirus, purchased from the Experimental Animal Center of Southern Medical University, were randomly divided into two groups, one of which was intramuscularly injected into the bilateral neck with $10^5$ $TCID_{50}$ of CSFV and the other with the same volume of phosphate-buffered saline (PBS) (42, 67). Before or after CSFV infection, the anterior vena cava blood of piglets was sterilely collected into heparin sodium anticoagulant tubes every other day, and peripheral blood mononuclear cells were isolated and subjected to flow cytometry analysis. Then $CD3^+$ T cells in PBMCs and splenic lymphocytes isolated at 7 d post-infection (dpi) were used for transcriptomic analysis. Piglets were euthanized at 7 dpi, immune tissues (tonsil, thymus, spleen, and lymph node) and non-immune tissues (lung and kidney) were collected and used for hematoxylin-eosin staining, quantitative reverse transcription polymerase chain reaction, Western blot, and IHC.

### Reagents and antibodies

The chemical reagents used in this study are as follows: dimethyl sulfoxide (Sigma-Aldrich, V900090), rapamycin (Cell Signaling Technology, 9904), bafilomycin A1 (MedChemExpress, HY-100558), carbonyl cyanide 3-chlorophenylhydrazone (Sigma-Aldrich, C2759), recombinant TNF-α protein (Abcam, ab9642), SM-164 (Selleck, S7089), Z-VAD-FMK (MedChemExpress, HY-16658B), and necrosulfonamide (MedChemExpress, HY-100573). Opti-MEM I reduced serum medium (31985088) and Lipofectamine 3000 transfection reagent (L3000015) were purchased from Thermo Fisher Scientific.

The primary antibodies used in this study were as follows: mouse monoclonal anti-TNF-α (Santa Cruz Biotechnology, sc-52746), mouse monoclonal anti-IL-1β (Santa Cruz Biotechnology, sc-32294), rabbit monoclonal anti-TRIM25 (Abcam, ab167154), rabbit polyclonal anti-RIPK1 (Beyotime Biotechnology, AF7896), mouse monoclonal anti-RIPK3 (Santa Cruz Biotechnology, sc-374639), rabbit monoclonal anti-MLKL (Santa Cruz Biotechnology, sc-293201), mouse monoclonal anti-ZBP1 (Santa Cruz Biotechnology, sc-166344), rabbit monoclonal anti-p62 (Cell Signaling Technology, 8025), rabbit polyclonal anti-LC3B (Cell Signaling Technology, 2775), rabbit monoclonal anti-VDAC1 (Beyotime Biotechnology, AF-1024), rabbit monoclonal anti-TOMM20 (Beyotime Biotechnology, AF-1717), rabbit polyclonal anti-COX IV (Beyotime Biotechnology, AC610), rabbit monoclonal anti-HSP60 (Beyotime Biotechnology, AG2237), mouse monoclonal anti-ubiquitin (Cell Signaling Technology, 3936), mouse monoclonal anti-CSFV E2 (Jai Balajee Trading Company, 9011), rabbit polyclonal anti-N[pro] (kindly donated by professor Xinglong Yu, Hunan Agricultural University, China), mouse monoclonal anti-GAPDH (Beyotime Biotechnology, AG019), and mouse monoclonal anti-Tubulin (Beyotime Biotechnology, AT819). The secondary antibodies used for immunofluorescence were Alexa Fluor 555 donkey anti-rabbit IgG (Beyotime Biotechnology, A0412), Alexa Fluor 488 goat anti-mouse IgG (Beyotime Biotechnology, A0428),

Dylight 405 goat anti-rabbit IgG (Beyotime Biotechnology, A0605), and Alexa Fluor 555 donkey anti-mouse IgG (Beyotime Biotechnology, A0460).

## Transcriptomic analysis

Total RNA was extracted from CD3+ T cells using Trizol reagent. The concentration and integrity of RNA were assessed by assays using K550 and Agilent 2200 Tapestation, respectively. After qualifying samples, eukaryotic mRNA was enriched using magnetic beads with Oligo (dT); after interrupting the mRNA, the mRNA was used as a template to construct a complementary DNA (cDNA) library. After the construction, the insert length and effective concentration of the library were checked to ensure the quality of the library. After the library examination was qualified, the different libraries were pooled according to the amount of under-target data for sequencing using the Illumina platform. To guarantee the quality of subsequent analysis, adaptor sequences need to be removed, and reads with low quality (low quality, bases with a base quality value less than or equal to 25 accounted for more than 60% of the whole reads) and $N$ ($N$ means base information cannot be determined) proportion greater than 5% were filtered out to obtain clean reads that could be used for subsequent analysis. Clean reads were sequence aligned to the indicated genome using HISAT2 software, obtaining information on their position on the reference genome.

Fragments per kilobase of exon model per million mapped reads were calculated for each gene in individual samples using FeatureCounts, a method that eliminates the impact of differences in length and sequencing on calculated gene expression, i.e., normalizing for gene expression. Differential expression analysis among genes was performed using DESeq2 software, and differentially expressed genes (DEGs) were screened with the criteria of |log2foldchange| >1 and $P_{adj}$ <0.05. The functional annotation of DEGs by gene oncology (http://www.geneontology.org) and the enrichment analysis by Kyoto Encyclopedia of Genes and Genomes (http://www.genome.jp/kegg/) pathway were performed using DESeq2 software with a hypergeometric test to discover the biological processes and signal transduction pathways that DEGs were involved in.

## qRT-PCR assay

The qRT-PCR assay used in this study was performed as previously described (42). Briefly, total RNA was extracted from the cells of each group using TRIzol reagent (Invitrogen, 15596026) or a total RNA Kit I (Omega, R6834-02). cDNA was synthesized using HiScript II Q RT SuperMix (Vazyme, R223-01) for detecting relative mRNA expression of the indicated genes. The qRT- PCR assay was performed using ChamQ Universal SYBR qPCR Master Mix (Vazyme, Q711-02) using an iQ5 iCycler detection system (Bio-Rad, 1855200). Relative mRNA expressions of the specific genes were determined using the $2^{-\Delta\Delta CT}$ method and normalized to the housekeeping gene β-actin.

## Flow cytometry assay

The collected PBMCs were washed in PBS and incubated for 15 min with FITC mouse anti-Pig CD3 (BD Biosciences, 559582) and PE mouse anti-Pig CD4a (BD Biosciences, 559586), according to the manufacturer's protocol. PK-15 cells after a viral infection or drug treatment were collected using trypsin-EDTA and washed in PBS. The cells were resuspended in annexin buffer and incubated for 15 min with 2.5 µg/mL PI (BD Biosciences) and Annexin V (BD Biosciences), according to the manufacturer's protocol. The antibody-labeled or necrotic cells were analyzed using the flow cytometer (Beckman Coulter Life Science, CytoFLEX). Flow cytometry analyses were reproduced by three independent experiments.

## Western blot analysis

Cells collected from each group were washed twice in cold PBS and then lysed on ice with cell lysis buffer (Beyotime Biotechnology, P0013B) containing 1 mM phenylmethyl-sulfonyl fluoride (PMSF) (Beyotime Biotechnology, ST506) for 20 min. After centrifugation at 13,000 g/min for 15 min at 4℃, the supernatant was obtained and subject to protein quantification with a BCA Protein Assay Kit (Beyotime Biotechnology, P0012) and then boiled for 10 min in 5× SDS PAGE sample loading buffer (Beyotime Biotechnology, P0015L). Equal protein samples were separated on 10% SDS-PAGE and transferred onto a polyvinylidene difluoride (PVDF) membrane. The PVDF membranes were first blocked in PBS containing 5% non-fat milk powder and 0.1% Tween 20 at 37℃ for 1 h, which were then incubated with primary antibodies at 4℃ overnight and then with the correspond-ing secondary antibodies conjugated to HRP at 37℃ for 2 h. The immunolabeled protein complexes were visualized using an ECL Plus Kit (Beyotime Biotechnology, P0018), using the CanoScan LiDE 100 scanner system (Canon).

The gray-scale value of the specific bands was analyzed with ImageJ software for the semi-quantitation of protein expression.

## Confocal immunofluorescence microscopy

Cells were grown and treated in 35 mm Petri dishes (NEST, GBD-35-20) with a glass bottom. Then, the cells were washed with PBS and fixed with 4% paraformaldehyde (Sigma-Aldrich, P6148) for 30 min at room temperature and then permeated with 0.1% Triton X-100 (Sigma-Aldrich, T8787) for 10 min, followed by staining of nuclei with DAPI (Beyotime Biotechnology, C1002). Fluorescence signals were observed using a TCS SP2 confocal fluorescence microscope (Leica TCS SP8).

## Statistical analysis

The data are shown as the mean ± standard deviation of at least three independent experiments and were analyzed by one-way ANOVA or two-way ANOVA tests using the GraphPad Prism 9. $P$-values of less than 0.05 were considered to be statistically significant.

## ACKNOWLEDGMENTS

This work was supported by The Program of National Natural Science Foundation of China (No. 32172824, No. 32102643, and No. 32002287), Guangdong Major Project of Basic and Applied Basic Research (No. 2020B0301030007), The Science and Technology Program of Guangzhou, China (No. 202206010161), and Quality and Efficiency Improve-ment Project of South China Agricultural University (No. C18).

## AUTHOR AFFILIATIONS

[1]College of Veterinary Medicine, South China Agricultural University, Guangzhou, China
[2]Agro-Biological Gene Research Center, Guangdong Academy of Agricultural Sciences, State Key Laboratory of Livestock and Poultry Breeding industry, Guangzhou, China
[3]Key Laboratory of Zoonosis Prevention and Control of Guangdong Province, South China Agricultural University, Guangzhou, China

## AUTHOR ORCIDs

Keke Wu  http://orcid.org/0009-0003-0342-9072
Jinding Chen  http://orcid.org/0000-0002-5435-0079

## FUNDING

| Funder | Grant(s) | Author(s) |
|---|---|---|
| MOST | National Natural Science Foundation of China (NSFC) | 32172824 | Jinding Chen |
| MOST | National Natural Science Foundation of China (NSFC) | 32102643 | Shuangqi Fan |
| MOST | National Natural Science Foundation of China (NSFC) | 32002287 | Xiaoai Zhang |

## AUTHOR CONTRIBUTIONS

Keke Wu, Conceptualization, Data curation, Formal analysis, Writing – original draft | Bingke Li, Conceptualization, Methodology | Xiaoai Zhang, Funding acquisition | Yiqi Fang, Conceptualization, Methodology | Sen Zeng, Software | Wenshuo Hu, Data curation, Resources | Xiaodi Liu, Validation | Xueyi Liu, Visualization | Zhimin Lu, Validation | Xiaowen Li, Project administration, Writing – review and editing | Wenxian Chen, Supervision | Yuwei Qin, Validation | Bolun Zhou, Methodology, Resources | Linke Zou, Methodology, Supervision | Feifan Zhao, Supervision | Lin Yi, Project administration | Mingqiu Zhao, Writing – review and editing | Shuangqi Fan, Project administration, Visualization, Writing – review and editing, Funding acquisition | Jinding Chen, Funding acquisition, Project administration, Writing – review and editing

## DATA AVAILABILITY

The data that support the findings of this study are openly available in figshare at http://doi.org/10.6084/m9.figshare.24721812.

## ETHICS APPROVAL

The authors declare that the animal experimental protocol was approved by the Animal Ethics Committee of South China Agricultural University [approval ID: SYXK(Yue）2019–0136]. All animal care and use protocols in this study were performed in accordance with the approved current guidelines.

## ADDITIONAL FILES

The following material is available online.

### Supplemental Material

**Supplemental material (Spectrum02758-23-s0001.pdf).** Figures S1 to S5.

### Open Peer Review

**PEER REVIEW HISTORY (review-history.pdf).** An accounting of the reviewer comments and feedback.

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
