## [Reviewer comments · Microbiology Spectrum]

Microbiology Spectrum

CSFV restricts necroptosis to sustain infection by inducing autophagy/mitophagy targeted degradation of RIPK3

Keke Wu, Bingke Li, Xiaoi Zhang, Yiqi Fang, Sen Zeng, Wenshuo Hu, Xiaodi Liu, Xueyi Liu, Zhimin Lu, Xiaowen Li, Wenxian Chen, Yuwei Qin, Bolun Zhou, Linke Zhou, Feifan Zhao, Lin Yi, Mingqiu Zhao, Shuangqi Fan, and Jinding Chen

Corresponding Author(s): Jinding Chen, South China Agricultural University College of Veterinary Medicine

Review Timeline:

Submission Date:	July 4, 2023
Editorial Decision:	August 7, 2023
Revision Received:	September 8, 2023
Editorial Decision:	October 13, 2023
Revision Received:	October 13, 2023
Accepted:	November 10, 2023

Editor: Leiliang Zhang

Reviewer(s): Disclosure of reviewer identity is with reference to reviewer comments included in decision letter(s). The following individuals involved in review of your submission have agreed to reveal their identity: Bin Zhou (Reviewer #2)

Transaction Report:

DOI: <https://doi.org/10.1128/spectrum.02758-23>

August 7, 2023

Prof. Jinding Chen
South China Agricultural University College of Veterinary Medicine
Wushan Street
Tianhe District
Guangzhou, Guangdong 510642
China

Re: Spectrum02758-23 (CSFV restricts necroptosis to sustain infection by inducing autophagy/mitophagy targeted degradation of RIPK3 via NS4A-TRIM25)

Dear Prof. Jinding Chen:

Link Not Available

Sincerely,

Leiliang Zhang

Journals Department
Reviewer comments:

Reviewer #1 (Comments for the Author):

In this manuscript Wu et al demonstrate that CSFV restricts necroptosis to sustain infection by inducing autophagy/mitophagy targeted degradation of RIPK3 via NS4A-TRIM25. They further show that CSFV NS4A protein promoted tripartite motif containing 25 (TRIM25) expression, synergistically induced the occurrence of mitophagy, targeted the autophagic degradation of RIPK3 to block the progression of necroptosis occurrence, and achieved persistent viral infection. These findings provide evidence supporting that the CSFV-induced autophagy pathway plays an important role in counteracting host cell necrosis, enriching our knowledge of pathogens that may subvert and evade necroptosis this host defense, and shed new light on

understanding the mechanisms of T lymphocyte exhaustion and immunosuppression during CSFV infection. While the biochemical and molecular experiments and characterization looks convincing and provides quite clear data, my main concern is the lack of experimental validation results of NSA4-TRIM25, NSA4-NDP52, NSA4-RIPK3 interactions under conditions of viral infection. Some clarification and additions are required. Some typos or grammatical errors need to be fixed.

Reviewer #2 (Comments for the Author):

This study demonstrates that necroptosis involves in the necrosis of T lymphocytes in the spleen and peripheral blood of pigs infected in vivo with CSFV. Meanwhile, the induction of autophagy by CSFV at a later stage of infection clearly restricts necroptosis in vitro. Then, it is revealed that CSFV NS4A protein promoted TRIM25 expression, synergistically induced the occurrence of mitophagy, targeted the autophagic degradation of RIPK3 to block the progression of necroptosis occurrence, and achieved persistent viral infection. These findings provide evidence supporting that the CSFV-induced autophagy pathway plays an important role in counteracting host cell necrosis, enriching our knowledge of pathogens that may subvert and evade necroptosis this host defense, and shed new light on understanding the mechanisms of T lymphocyte exhaustion and immunosuppression during CSFV infection.

However, some results of this text are of low quality, and the interpretation of some results lack logic, as follows:

Major points

1. In Figure 3I, J, K and L, only the changes of RIPK3, TNF α , TNFR1 and ZBP1 mRNA levels were detected. Supplementary experiments are recommended to detect its changes at the protein level.
2. In Figure 3I, K, J and L, the mRNA levels of RIPK1, RIPK3, and TNF- α in PBMC increased significantly after CSFV infection, but the mRNA levels of RIPK1, RIPK3, and TNF- α in Spleen did not change significantly. It is suggested to explain with corresponding sentences in the results or discussion section. In addition, please explain why the results of immunohistochemistry in Figure 3M-O show that the expression of RIPK1 in the spleen is increased, which is not consistent with the results of qPCR.
3. In Figure 4A, B and 4G, why were the changes of RIPK1, RIPK3 and ZBP1 at protein and mRNA levels inconsistent at 48 hpi and 60 hpi? Please reconfirm and explain.
4. The confocal images such as Figure 5B, 8C, 8D, 8E and 9F are overexposed, please replace them with higher quality images.
5. The images of Western blot such as Figure 5B, 5E, 5F, 5H, 5I, Figure S3 were not subjected to gray-scale analysis. Please add the results of grayscale analysis.
6. In Figure S2A, the image quality of CO-IP result is too low, please replace it. In Figure S2B, pEGFP-C1 is not expressed, and it cannot be proved from this figure that there is no co-localization of pEGFP-C1 and My-TRIM25, and this negative control does not hold. Therefore, Figure S2A cannot prove that My-TRIM25 interacts specifically with GFP-NS4A. In addition, the same problem exists in Figure 5G.
7. In Figure S3, why the expression of LC3II was down-regulated after Rapamycin and CCCP treatment. Please confirm and explain.
8. Necrosis-related genes were up-regulated in both PBMC and spleen at 7 dpi in vivo study, but down-regulated at 60 dpi in vitro study. There is a contradiction between them. Please provide a reasonable explanation in the results or discussion section.

Minor points

1. In Figure 1A, the fluorescent yellow font used is not clear, and it is recommended to replace it with other clearer colors for the convenience of readers.
2. Figure 1B is the result of multiple data analyses, while Figure 1A seems to show only one set of data, it is recommended to show the whole data. If only the most typical data are shown, please indicate in the figure legend.
3. In Figure 4G, RIPK1 mRNA levels were not significantly different between CSFV-infected cells and MOCK cells. However, lines 253-255 describe that "CSFV could obviously increase RIPK1, RIPK3 and ZBP1 mRNA levels in PBMCs compared with the control", This is not consistent with the results. The language is not rigorous enough. Please amend it.
4. In line 279, it should be Figure S2B, not Figure 2B. Please amend it.
5. In Figure 5E, is RIPK3 antibody or HA antibody used to detect RIPK3? If HA antibody was used, please mark it clearly.

Staff Comments:

Preparing Revision Guidelines

Please return the manuscript within 60 days; if you cannot complete the modification within this time period, please contact me. If you do not wish to modify the manuscript and prefer to submit it to another journal, please notify me of your decision immediately so that the manuscript may be formally withdrawn from consideration by Microbiology Spectrum.

Reviewer #1: In this manuscript Wu et al demonstrate that CSFV restricts necroptosis to sustain infection by inducing autophagy/mitophagy targeted degradation of RIPK3 via NS4A-TRIM25. They further show that CSFV NS4A protein promoted tripartite motif containing 25 (TRIM25) expression, synergistically induced the occurrence of mitophagy, targeted the autophagic degradation of RIPK3 to block the progression of necroptosis occurrence, and achieved persistent viral infection. These findings provide evidence supporting that the CSFV-induced autophagy pathway plays an important role in counteracting host cell necrosis, enriching our knowledge of pathogens that may subvert and evade necroptosis this host defense, and shed new light on understanding the mechanisms of T lymphocyte exhaustion and immunosuppression during CSFV infection.

While the biochemical and molecular experiments and characterization looks convincing and provides quite clear data, my main concern is the lack of Lack of experimental validation results of NSA4-TRIM25, NSA4-NDP52, NS4A-RIPK3 interactions under conditions of viral infection. Some clarification and additions are required. Some typos or grammatical errors need to be fixed.

Thus there are some major concerns

1. Mitochondria autophagy (mitophagy) and macroautophagy (the most prevalent form of autophagy) differ significantly in terms of regulatory mechanisms, target substances, selectivity and effects. In macroautophagy, cells can choose to break down and remove any unwanted or harmful cellular components. For example, intracellular protein aggregates, excessive mitochondria or endoplasmic reticulum can be the target substances of autophagy. Mitophagy, on the other hand, specifically targets damaged or excessive mitochondria within the cell. Mitophagy is initiated only when mitochondria become dysfunctional or otherwise damaged. Macroautophagy is regulated by the involvement of the ATG protein family, whereas mitochondrial autophagy is responsible for the regulation by a specific protein family related to mitochondria, e.g., the proteins PARKIN and PINK1 play a key role in regulating the delivery process of mitochondrial autophagy after mitochondrial damage. PINK1 is the specific marker of mitophagy, which regulates mitophagy by changing the membrane potential of mitochondria. Therefore, it is recommended to increase the WB detection of PINK1 protein to determine that CSFV and NS4A activate mitophagy rather than macroautophagy in 7B,E,G. Please reconsider the title of this paper should be autophagy or mitophagy.
2. In Figure 3A, c, f, h, please add the full zoom line, it looks like there is a dotted line missing from each, and which of the graphs 3B and C results in statistics 3A, and which of the graphs 3E and F results in statistics 3D, please label.
3. The magnification should be mentioned in the figure legends as the magnification

of Mock and CSFV in figures 3M and 3N seem to be different.

4. In Figure 3G, lane 1-3 and lane 4-6 represent three replicates or different time points, which is not stated in the text, and the expression level of MLKL protein is gradually decreasing in 4-6 and not gradually increasing as described in the text (Page 9 line 215), which suggests that a analysis of the gray intensity of the bands be performed to show more accurate results.
5. NS4A protein overexpression did not lead to a decrease in the protein level of RIPK3 in Figure 5B, and the overexpression of NS4A did not attenuate the red fluorescence of PIPK3 labeling in Figure 5A, please perform the grayscale analysis and make sure that the results of fluorescence confocal experiments under the same exposure intensity.
6. In Figure 5B, the different expression levels of P7,C,NS4A,NS4B, and NS5A protein may lead to different results on the expression of PIPK3 and MLKL. It is suggested that the expression levels of P7,C,NS4A,NS4B, and NS5A should be controlled at a consistent level by adjusting the amount of each transfected expression plasmid to re-evaluate the influence on PIPK3 and MLKL expression.
7. Line 279 should be Figure S2B not Figure 2B.
8. The superscript GFP-NS4A in Figure S2B should be changed to GFP, and there should be green fluorescence in the Myc-TRIM25+PeGFP-C1 group because empty GFP also shows green, suggesting a reconsideration of the figure selection. The antibodies used should be specified for Myc-TRIM25 as well, Myc or TRIM25? Includes all other confocal fluorescence images.
9. I suggest a pull-down assay to demonstrate that interaction between NS4A-TRIM25 and NS4A-RIPK3.
10. In Figure 5E, co-transfection of HA-RIPK3 and GFP-NS4A plasmids detected the effect of NS4A on the expression of RIPK3. HA antibody should be used to detect PIPK3, not PIPK3 antibody, because the total RIPK3 protein, including overexpressed and endogenous, is detected with RIPK3 antibody. In addition, additional experiments are needed to transfect only NS4A plasmid to explore the effect of NS4A overexpression on endogenous RIPK3 expression. Figure 5F shows the same situation, need to add endogenous experiments
11. Figure 5G images are blurred and the fluorescence intensity profiles cannot show their interaction (especially RIPK3 and NS4A, NS4A and TRIM25), so it is recommended to replace them with clearer images for re-analysis. Meanwhile, it is suggested to supplement the fluorescence confocal experiments on the interaction of NS4A with endogenous TRIM25 and PIPK3 in the case of viral

infection. A line profile should be added indicating the position where the fluorescence intensities has been measured.

12. In Figure 5I, the expression of endogenous TRIM25 should be detected with TRIM25 antibody to ensure that TRIM25 were knockdown, and also should be provided a gray intensity analysis of Western blot.
13. A supplementary experiment was conducted to study the effect of TRIM25 knockdown on CSFV replication.
14. The level of autophagy was not activated in normal growing cells, and the expression of LC3-I was higher than that of LC3-II. In Figure 6A, the amount of LC3-II in DMSO-treated cells was much higher than LC3-I, indicating that autophagy was activated, which was contrary to the general cognition, and it was suggested to consider repeating the experimental results. Moreover, in Figure 6B, LC3-II accumulation was not observed in the DMSO treated group under the same condition of PK15 cells. What is the reason for this discrepancy?
15. Figure S3 legend: "(A)" should perhaps go at the start of this paragraph.
16. Mitochondria localized to lysosomes appear red, rather than yellow due to the acidic quenching of GFP and stability of mCherry in the low pH environment. From Figure 7J-I, it can be seen that the NS4A protein induced incomplete mitophagy rather than complete mitophagy as described by the authors.
17. Authors should unify the use of LC3-I and LC3-II throughout the text, includes all figures.
18. There are no data describing the effect of NDP52 knockdown on the degradation of PIPK3 protein and CSFV replication.
19. Line 113: The format for references is in need of revision. Please change "33343536373839" to "33-39" .
20. I strongly recommend that authors should find someone native in English to revise this manuscript for unneglected grammatical errors.

Response to Reviewer 1 Comments

Dear Editors and Reviewers:

Thank you for your letter and for the reviewers' comments concerning our manuscript entitled "CSFV restricts necroptosis to sustain infection by inducing autophagy/mitophagy targeted degradation of RIPK3 via NS4A-TRIM25" (ID: Spectrum02758-23). Those comments are all valuable and very helpful for revising and improving our paper, as well as the important guiding significance to our researches. We have studied comments carefully and have made correction which we hope meet with approval. Revised portion are marked in red with track change mode in the paper. The main corrections in the paper and the responds to the reviewer's comments are as follows:

Point 1: My main concern is the lack of experimental validation results of NS4A-TRIM25, NS4A-NDP52, NS4A-RIPK3 interactions under conditions of viral infection. Some clarification and additions are required.

Response 1: We sincerely apologize for our negligence regarding the evidence of the interaction between NS4A and the target protein under CSFV infection conditions. We fully agree that supplementing the experimental details will help strengthen the convincing nature of the interaction involving CSFV NS4A protein. However, due to the current limitations in experimental conditions (mainly the lack of CSFV NS4A antibodies), as shown in Figure S3, we have only been able to supplement the co-transfection of eukaryotic expression plasmids and simultaneous CSFV infection, followed by confocal microscopy to observe co-localization. This attempt aims to simulate the interaction under CSFV infection conditions. In future work, we hope to obtain CSFV NS4A antibodies and further refine the validation of NS4A interaction under CSFV infection conditions.

Point 2: Some typos or grammatical errors need to be fixed.

Response 2: We are very sorry for our inaccurate description especially the grammatical errors and typos. As Reviewer suggested, we have tried our best to improve the manuscript and made a thorough revision regarding the use of English language and grammar in the manuscript. These changes will not influence the content and framework of the paper. And here we have not listed all of the changes but marked in red in revised paper.

We appreciate for your warm work earnestly, and hope that the correction will meet with approval.

Once again, thank you very much for your comments and suggestions.

Response to Reviewer 2 Comments

Dear Editors and Reviewers:

Thank you for your letter and for the reviewers' comments concerning our manuscript entitled "CSFV restricts necroptosis to sustain infection by inducing autophagy/mitophagy targeted degradation of RIPK3 via NS4A-TRIM25" (ID: Spectrum02758-23). Those comments are all valuable and very helpful for revising and improving our paper, as well as the important guiding significance to our researches. We have studied comments carefully and have made correction which we hope meet with approval. Revised portion are marked in red with track change mode in the paper. The main corrections in the paper and the responds to the reviewer's comments are as follows:

Major points

Point 1: In Figure 3I, J, K and L, only the changes of RIPK3, TNF α , TNFR1 and ZBP1 mRNA levels were detected. Supplementary experiments are recommended to detect its changes at the protein level.

Response 1: We deeply apologize for the lack of evidence regarding the protein levels of RIPK3, TNF α , TNFR1, and ZBP1. We fully agree that supplementing this experiment will help to improve the details of the regulation of Necroptosis-related protein levels during CSFV infection. However, considering that the PBMC/spleen samples from the animal experiments have been stored for nearly two years, it may not be accurate to add this experimental data at this point.

But as shown in Figure 3G, we have examined the protein levels of RIPK1, MLKL, TNF α , and IL-1 β in CSFV-infected PBMCs *in vivo*. Although there is a lack of indicators for RIPK3 and ZBP1, it can still reflect to some extent the regulation of protein levels of Necroptosis-related molecules in CSFV-infected PBMCs. In addition, as shown in Figures 3A and H, we have also analyzed the cell morphology and clustering characteristics that correspond to Necroptosis induced by CSFV infection in PBMCs using transmission electron microscopy and flow cytometry, respectively.

As for the spleen samples, as shown in Figure 3N, we primarily used immunohistochemistry to visually label key molecules instead of detecting protein levels.

Point 2: In Figure 3I, K, J and L, the mRNA levels of RIPK1, RIPK3, and TNF- α in PBMC increased significantly after CSFV infection, but the mRNA levels of RIPK1, RIPK3, and TNF- α in Spleen did not change significantly. It is suggested to explain with corresponding sentences in the results or discussion section. In addition, please explain why the results of immunohistochemistry in Figure 3M-O show that the expression of RIPK1 in the spleen is increased, which is not consistent with the results of qPCR.

Response 2: We deeply apologize for the lack of accurate description and explanation regarding the changes in the regulation levels of RIPK1, RIPK3, and TNF- α in the spleen. We have added an explanation for these results in lines 233-234, stating, "This may be attributed to significant differences among the RNA samples within the group." Furthermore, due to the inability to determine the regulation trends of RIPK1 and RIPK3 at the mRNA level, we have supplemented the study by using immunohistochemistry to further determine the regulation levels of key molecules.

Point 3: In Figure 4A, B and 4G, why were the changes of RIPK1, RIPK3 and ZBP1 at protein and mRNA levels inconsistent at 48 hpi and 60 hpi? Please reconfirm and explain.

Response 3: We deeply apologize for the oversight in explaining the inconsistency between the mRNA and protein levels of the relevant molecules in Figure 4A, B, and 4G. We have re-evaluated this result and provided an explanation in lines 261-264, stating, "This observation may appear contradictory to the results at the protein level but is consistent with the conclusion that CSFV infection can significantly induce necrotic apoptosis in PBMCs. As for the protein level inhibition observed in the later stages, it is likely closely related to the autophagy process induced by CSFV."

We acknowledge that the mRNA and protein levels may not always exhibit a direct correlation, and this discrepancy could be attributed to various factors, including post-transcriptional modifications and protein degradation processes. In the case of CSFV infection-induced necroptosis, the modulation of mRNA and protein levels might follow different regulatory mechanisms.

Furthermore, the observed protein level inhibition in the later stages of CSFV infection suggests the involvement of the autophagy process. Autophagy can play a role in regulating protein degradation and turnover, which could explain the observed decrease in protein levels during later stages of CSFV infection.

We apologize for any confusion caused by the initial oversight and appreciate the opportunity to provide a more comprehensive explanation of the results.

Point 4: The confocal images such as Figure 5B, 8C, 8D, 8E and 9F are overexposed, please replace them with higher quality images.

Response 4: We apologize for the low quality of Figure 5B, 8C, 8D, 8E, and 9F confocal microscopy images. We have now supplemented the relevant experiments or replaced them with higher quality images to enhance the credibility and persuasiveness of our research.

Point 5: The images of Western blot such as Figure 5B, 5E, 5F, 5H, 5I, Figure S3 were not subjected to gray-scale analysis. Please add the results of grayscale analysis.

Response 5: We apologize for the lack of grayscale value analysis in Figure 5B, 5E, 5F, 5H, 5I, and Figure S3. We fully agree that supplementing the grayscale value analysis for these results will provide clearer logic and organization. Following your suggestion, we have now added the grayscale value analysis for the relevant results in Figure 5 and Figure S2. We have also made the necessary revisions to the figure captions by adding and modifying them in the appropriate locations using the track changes mode.

Point 6: In Figure S2A, the image quality of CO-IP result is too low, please replace it. In Figure S2B, pEGFP-C1 is not expressed, and it cannot be proved from this figure that there is no co-localization of pEGFP-C1 and My-TRIM25, and this negative control does not hold. Therefore, Figure S2A cannot prove that My-TRIM25 interacts specifically with GFP-NS4A. In addition, the same problem exists in Figure 5G.

Response 6: Thank you very much for your valuable suggestions. We apologize for the low quality of the images and the lack of an effective pEGFP-C1 control. As shown in Figure S2 and Figure 5, In response to your advice, we have supplemented the relevant experiments or replaced the images with more accurate and higher quality ones to support our findings and conclusions.

Point 7: In Figure S3, why the expression of LC3 II was down-regulated after Rapamycin and CCCP treatment. Please confirm and explain.

Response 7: Thank you very much for your comments and suggestions. We apologize for providing controversial and low-quality results. Considering your advice and the logical interpretation of the results, we have ultimately decided to retain only the uncontroversial validation results conducted on PK-15 cells.

Point 8: Necrosis-related genes were up-regulated in both PBMC and spleen at 7 dpi in vivo study, but down-regulated at 60 dpi in vitro study. There is a contradiction between them. Please provide a reasonable explanation in the results or discussion section.

Response 8: Thank you very much for your comments and suggestions. We sincerely apologize for providing controversial descriptions. Your input has been invaluable in revising our manuscript. Based on your advice, we have provided a more detailed explanation of these results in the revised manuscript, specifically in the discussion section from lines 550 to 561.

“CSFV induces necroptosis in PBMCs and spleen in vivo, but inhibits necroptosis levels in porcine kidney-derived cell line (PK-15) and alveolar macrophage cell line (3D4/21) in vitro at 60 dpi. On one hand, the induction of necroptosis is closely related to the levels of autophagy induced during CSFV infection, suggesting that autophagy induction may precede necroptosis, ultimately leading to the suppression of necroptosis levels. On the other hand, CSFV infection displays different tissue tropism and induces varying degrees of pathology in different tissues. For example, it can cause splenic marginal hemorrhage and infarction, as well as dense subcapsular petechiae in the kidneys. However, CSFV infection in vitro does not induce cellular pathology. This suggests that the activation/inhibition of necroptosis by CSFV in vivo and in vitro is of great relevance to the corresponding tissue/target cell types.”

We hope this revision is acceptable to you and provides a better explanation of our research findings.

Minor points

Point 1: In Figure 1A, the fluorescent yellow font used is not clear, and it is recommended to replace it with other clearer colors for the convenience of readers.

Response 1: Thank you very much for your comments. Based on your suggestions, we have changed the originally inconvenient fluorescent yellow font and boxes to a darker shade of yellow to improve readability.

Point 2: Figure 1B is the result of multiple data analyses, while Figure 1A seems to show only one set of data, it is recommended to show the whole data. If only the most typical data are shown, please indicate in the figure legend.

Response 2: Thank you very much for your feedback. We acknowledge that Figure 1A only displays data from one set of results in Figure 1B. Due to limited space in the figure, we have followed your suggestion and indicated in the figure caption that Figure 1A represents a representative image, aiming to enhance the rigor of the results. We hope this modification meets your requirements and ensures the accuracy of the presented results.

Point 3: In Figure 4G, RIPK1 mRNA levels were not significantly different between CSFV-infected cells and MOCK cells. However, lines 253-255 describe that "CSFV could obviously increase RIPK1, RIPK3 and ZBP1 mRNA levels in PBMCs compared with the control", This is not consistent with the results. The language is not rigorous enough. Please amend it.

Response 3: We apologize for the inaccuracies in the description of this result. The statement "CSFV could obviously increase RIPK1, RIPK3, and ZBP1 mRNA levels in PBMCs compared with the control" has been modified to "CSFV could obviously increase RIPK3 and ZBP1 mRNA levels in PBMCs compared with the control."

Point 4: In line 279, it should be Figure S2B, not Figure 2B. Please amend it.

Response4: We regret any confusion caused by the previous description and appreciate your attention to detail. In the revised manuscript, we have made the necessary correction.

Point 5: In Figure 5E, is RIPK3 antibody or HA antibody used to detect RIPK3? If HA antibody was used, please mark it clearly.

Response5: Thank you for your attention to the details of the manuscript. We have reviewed the information regarding the detection of RIPK3 in Figure 5E (Figure S2C in revised manuscript) once again, and we can confirm that RIPK3 was indeed detected using an RIPK3 antibody rather than an HA tag antibody.

We appreciate for your warm work earnestly, and hope that the correction will meet with approval.

Once again, thank you very much for your comments and suggestions.

October 13, 2023

Prof. Jinding Chen
South China Agricultural University College of Veterinary Medicine
Wushan Street
Tianhe District
Guangzhou, Guangdong 510642
China

Re: Spectrum02758-23R1 (CSFV restricts necroptosis to sustain infection by inducing autophagy/mitophagy targeted degradation of RIPK3)

Dear Prof. Jinding Chen:

Thank you for submitting your manuscript to Microbiology Spectrum. As you will see your paper is very close to acceptance. Please modify the manuscript along the lines I have recommended. As these revisions are quite minor, I expect that you should be able to turn in the revised paper in less than 30 days, if not sooner. If your manuscript was reviewed, you will find the reviewers' comments below.

When submitting the revised version of your paper, please provide (1) point-by-point responses to the issues raised by the reviewers as file type "Response to Reviewers," not in your cover letter, and (2) a PDF file that indicates the changes from the original submission (by highlighting or underlining the changes) as file type "Marked Up Manuscript - For Review Only". Please use this link to submit your revised manuscript. Detailed instructions on submitting your revised paper are below.

Link Not Available

Sincerely,

Leiliang Zhang

Reviewer comments:

Reviewer #1 (Comments for the Author):

The authors have addressed all the questions I have.
The manuscript needs to be edited for English grammar and writing conventions.

Reviewer #2 (Comments for the Author):

There are some minor writing issues in the manuscript that need to be addressed. In line 1277, the "50" in "TCID50" should be modified to a subscript.

Preparing Revision Guidelines

Please return the manuscript within 60 days; if you cannot complete the modification within this time period, please contact me. If you do not wish to modify the manuscript and prefer to submit it to another journal, please notify me of your decision immediately so that the manuscript may be formally withdrawn from consideration by Microbiology Spectrum.

Dear Editors and Reviewers:

Thank you for your letter and for the reviewers' comments concerning our manuscript entitled "CSFV restricts necroptosis to sustain infection by inducing autophagy/mitophagy targeted degradation of RIPK3" (ID: Spectrum02758-23). Those comments are all valuable and very helpful for revising and improving our paper, as well as the important guiding significance to our researches. We have studied comments carefully and have made correction which we hope meet with approval. Revised portion are marked in red with track change mode in the paper. The main corrections in the paper and the responds to the reviewer's comments are as follows:

Reviewer #1 (Comments for the Author):

The authors have addressed all the questions I have.
The manuscript needs to be edited for English grammar and writing conventions.

Response to Reviewer #1 Comments

As Reviewer suggested, we have tried our best to improve the manuscript and made a thorough revision regarding the use of English grammar and writing conventions in the manuscript. These changes will not influence the content and framework of the paper. And here we have not listed all of the changes but marked in red in revised paper.

Reviewer #2 (Comments for the Author):

There are some minor writing issues in the manuscript that need to be addressed. In line 1277, the "50" in "TCID50" should be modified to a subscript.

Response to Reviewer #2 Comments

We sincerely apologize for our negligence regarding the handwriting mistakes, we have corrected this error in line 1249 of the revised draft, and have carefully read through the entire text to revise some others writing issues in the revised manuscript.

We appreciate for your warm work earnestly, and hope that the correction will meet with approval.

Once again, thank you very much for your comments and suggestions.

Re: Spectrum02758-23R2 (CSFV restricts necroptosis to sustain infection by inducing autophagy/mitophagy targeted degradation of RIPK3)

Dear Prof. Jinding Chen:

Your manuscript has been accepted, and I am forwarding it to the ASM production staff for publication. Your paper will first be checked to make sure all elements meet the technical requirements. ASM staff will contact you if anything needs to be revised before copyediting and production can begin. Otherwise, you will be notified when your proofs are ready to be viewed.

Sincerely,
Leiliang Zhang
Editor
Microbiology Spectrum

Reviewer #2 (Comments for the Author):

The current manuscript status is significantly better than the former.